JGP Journal of General Physiology

# The calcium-binding protein S100A1 binds to titin's N2A insertion sequence in a pH-dependent manner

Sabrina I. Apel[1,2] [ID], Emily Schaffter[1,2] [ID], Nicholas Melisi[1,2] [ID], and Matthew J. Gage[1,2] [ID]

**Titin is the third contractile filament in the sarcomere, and it plays a critical role in sarcomere integrity and both passive and active tension. Unlike the thick and thin filaments, which are polymers of myosin and actin, respectively, titin is a single protein that spans from Z-disk to M-line. The N2A region within titin has been identified as a signaling hub for the muscle and is shown to be involved in multiple interactions. The insertion sequence (UN2A) within the N2A region was predicted as a potential binding site for the Ca²⁺-binding protein, S100A1. We demonstrate using a combination of size exclusion chromatography, surface plasmon resonance, and fluorescence resonance energy transfer that S100A1 can bind to the UN2A region. We further demonstrate that this interaction occurs under conditions where calcium is bound to S100A1, suggesting that the conformational shift in S100A1 when calcium binds is important. We also observed a conformational change in UN2A induced by shifts in pH, suggesting that conformational flexibility in UN2A plays a critical role in the interaction with S100A1. These results lead us to propose that the interaction of S100A1 and UN2A might act as a sensor to regulate titin's function in response to physiological changes in the muscle.**

## Introduction

Titin is the largest known protein spanning half a sarcomere from the Z-disk to the M-line in the striated muscle (Bang et al., 2001). Titin has been shown to have crucial roles in contributing to the enhancement of passive tension to the muscle (Linke et al., 1998), maintenance of sarcomeric structural integrity (Horowits and Podolsky, 1987), and is responsible for elements of the viscoelastic behavior of myofibrils and stiffness of the muscle (Nishikawa et al., 2020; Trinick, 1996; Gregorio et al., 1999). The titin gene (*Ttn*) consists of 363 exons and undergoes alternative splicing to express several different isoforms, specifically the N2A isoform in the skeletal muscle and the N2BA isoform in the cardiac muscle (Bang et al., 2001; Nishikawa et al., 2020; Guo et al., 2010). Mutations in the *Ttn* gene can result in titinopathies, inherited diseases of cardiac and skeletal muscle, and most of the disease-associated variants involve mutations resulting in significant changes to the expressed titin protein (Nishikawa et al., 2020; Savarese et al., 2016). Mutations in titin and their underlying mechanisms have yet to be fully elucidated and are critical for understanding titin's contribution to muscular diseases and physiological function.

Titin is comprised of two distinct regions: the N-terminal I-band that is composed of two regions of Ig domains and the disordered PEVK segment; and the C-terminal A-band region, composed of a mixture of Ig and fibronectin domains (Tskhovrebova and Trinick, 2010). The N2A region is unique to skeletal titin and the N2BA isoform of cardiac titin and is between the proximal tandem Ig region and the PEVK region in the I-band. Four Ig domains, I80, I81, I82, and I83, and a unique insertion sequence (UN2A) that is about 100 amino acids, residing between I80 and I81, comprise the N2A region (Fig. 1 A [Miller et al., 2003]). UN2A contains a disordered N-terminal and C-terminal with an ordered tri-helical structure in the center (Fig. 1, A and B [Tiffany et al., 2017; Stronczek et al., 2021; Zhou et al., 2016; Stehle et al., 2023]).

The N2A region plays a major role as a signaling hub for a number of protein–protein interactions (Nishikawa et al., 2020). Among the proteins known to bind in the UN2A region are the muscle ankyrin repeat proteins (MARPs [Miller et al., 2003]) and the calcium (Ca²⁺)-regulated cysteine protease (p94/calpain 3 [Miller et al., 2003; Sorimachi et al., 1995; Hayashi et al., 2008]), and it has been proposed that binding of these proteins to the N2A region help localize these proteins to titin for future signaling events (Miller et al., 2003). MARP1 was shown to facilitate N2A binding to F-actin, causing an increase in passive force and constraining overstretching of sarcomeres. (Van Der Pijl et al., 2021; Zhou et al., 2021) The p94/calpain-3–N2A interaction plays a role in stabilizing p94 by suppressing its autolysis, and ultimately, protecting titin from proteolysis (Hayashi et al., 2008; Ono et al., 2006). Understanding the specifics of the different interactions that occur within the N2A

---

[1]Chemistry Department, University of Massachusetts Lowell, Lowell, MA, USA; [2]UMass Movement Center, University of Massachusetts Lowell, Lowell, MA, USA.

Correspondence to Matthew J. Gage: matthew_gage@uml.edu.

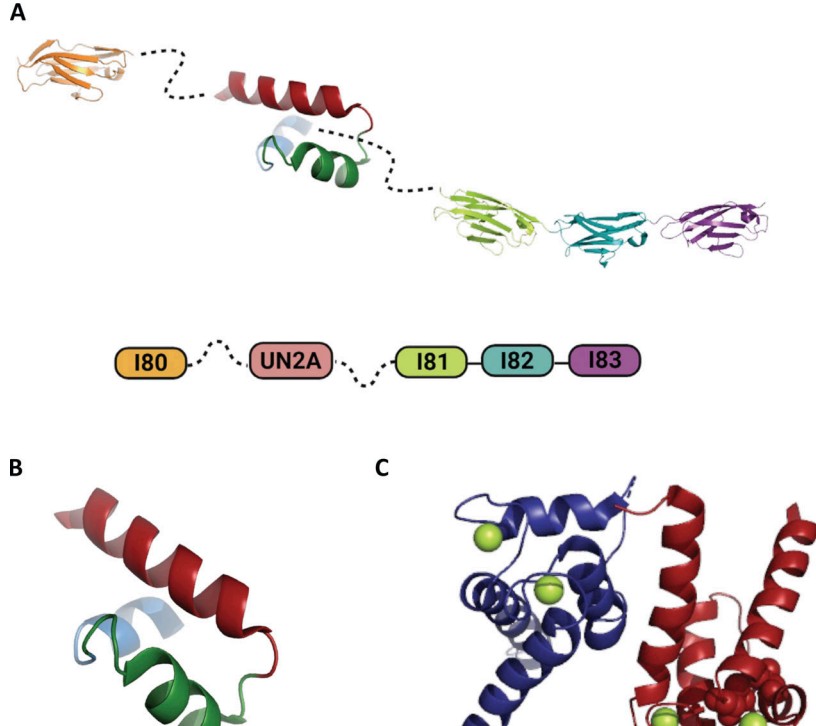

**A**

**B** **C**

Figure 1. **Structure and Organization of the N2A region and S100A1. (A)** Schematic of the structural domains in the N2A region. The N2A region consists of four Ig domains (I80, I81, I82, and I83) and a unique insertion sequence (UN2A) with a tri-helix bundle at its core and disordered termini (dashed lines). **(B)** Schematic of the tri-helix bundle in UN2A. **(C)** Schematic of the Ca²⁺-binding protein, S100A1. S100A1 exists as a homodimer in its native state and binds to four Ca²⁺ ions (green spheres).

region, like those with MARPs and p94, will provide greater insights into the role that the N2A region plays in the signaling pathways of titin, and muscles in general. An interaction that our lab has investigated is binding of N2A to actin filaments with $Ca^{2+}$ enhancing this binding interaction (Dutta et al., 2018). While investigating how $Ca^{2+}$ could enhance N2A/F-actin binding, we identified a potential binding site for S100A1 in the N2A region. S100A1 has previously been shown to bind titin's PEVK region so there was already an experimental evidence for S100A1 modulating titin activity (Fukushima et al., 2010; Gutierrez-Cruz et al., 2001; Yamasaki et al., 2001).

S100A1 is an important $Ca^{2+}$-binding protein that is thought to mediate regulatory $Ca^{2+}$ signaling pathways through specific interactions with its target proteins (Ranty et al., 2006; Kiewitz et al., 2000). $Ca^{2+}$ binding induces a conformational change that exposes binding sites for target proteins on the S100A1 (Sun and Kekenes-Huskey, 2020; Tang et al., 2020; Tidow and Nissen, 2013). There are a total of four $Ca^{2+}$-binding sites on S100A1 that have the canonical EF-hand binding motif, with two binding sites existing on each of monomer of the S100A1 dimer (Fig. 1 C [Sun and Kekenes-Huskey, 2020]). S100A1 is expressed in both the skeletal and cardiac muscle, but is generally found in higher concentrations in cardiac muscles compared with skeletal (Kato et al., 1986; Völkers et al., 2010; Most et al., 2003).

S100A1 is critical for regulating function of the RYR, so it has a critical role in regulating the influx of $Ca^{2+}$ from the sarcoplasmic reticulum (Prosser et al., 2011; Wright et al., 2008; Treves et al., 1997; Xu et al., 2017). The RYR is in an active state when another $Ca^{2+}$-binding protein, calmodulin (CaM), binds the receptor in the $Ca^{2+}$-free state and becomes inactivated when

CaM binds $Ca^{2+}$ and undergoes a conformational change (Woll et al., 2021; Rodney et al., 2000). S100A1 binds to RYR in the $Ca^{2+}$-bound state (Prosser et al., 2011) to strengthen the RYR's open probability upon reconstitution in the planar lipid bilayer (Treves et al., 1997) and this binding is thought to overlap with the CaM-binding site (Prosser et al., 2008; Ivanenkov et al., 1995). With this competition, it was proposed that the two $Ca^{2+}$-binding proteins compete for binding when there is an elevated $Ca^{2+}$ concentration to either activate or inactivate the channel through the displacement of its competitor (Wright et al., 2008). A study utilizing S100A1 knockout mice showed the importance of S100A1 binding in assisting with $Ca^{2+}$ release via the RYR (Prosser et al., 2008).

Our work on N2A and F-actin binding suggests that the $Ca^{2+}$ enhancement of binding appears to potentially be due to its stabilization of I83 (Kelly et al., 2021). However, in exploring potential mechanisms for this regulation, we hypothesized that since there is a putative S100A1 binding site in N2A, S100A1 might regulate N2A binding, similar to how it regulates binding of PEVK to F-actin (Yamasaki et al., 2001). In this work, we demonstrate that S100A1 can interact with UN2A, validating the putative binding site we identified. This binding occurs in the micromolar range, and the strength of the binding is regulated by both $Ca^{2+}$ concentration and pH. We further demonstrate that UN2A undergoes a conformational shift as pH decreases, which we hypothesize contributes to the increased binding at lower pHs since a similar shift is observed with binding of S100A1. Based on our results, we propose that S100A1 binding to titin helps tune muscle function in response to the changing physiological conditions of the sarcomere.

# Materials and methods

## Reagents

All the chemicals used were purchased from standard chemical suppliers, such as Fisher Scientific and Sigma-Aldrich. The sequence of the UN2A region was commercially synthesized by GeneArt (Invitrogen) and encompasses amino acids 8539–8655 in the mouse titin sequence (NCBI accession number: NM_011652.3), which shares 95% identity with the human UN2A sequence (111 out of the 117 amino acids are identical). The mouse *S100A1* gene sequence was graciously provided by David J. Weber's lab from the University of Maryland (College Point, MD, USA; NCBI accession number: NM_011309.3).

## Expression and purification of UN2A

The UN2A plasmid was transformed into BL21(DE3) pLysS cells and grown overnight. The overnight culture was inoculated into fresh LB media + ampicillin (100 µg/ml) + chloramphenicol (34 µg/ml) and was incubated at 37°C until $OD_{600}$ ~0.5 was reached. The culture was then induced with 1 mM IPTG and incubated at 18°C for 16–18 h. The overnight culture was harvested and resuspended in Anion Exchange (AEX) wash buffer (50 mM Tris, pH 7.5) containing lysozyme (250 µg/ml), DNAse (40 µg/ml), $MgCl_2$ (10 mM), and Triton X-100 (1% vol/vol). The resuspended pellet was lysed using a combination of freeze–thaw and sonication. Insoluble proteins were removed via centrifugation at 15,000 × *g*, and the resulting supernatant was loaded onto a Capto-Q AEX column (Cytiva) equilibrated with the wash buffer. The UN2A protein was eluted using a 0–100% gradient of elution buffer (50 mM Tris and 1 M NaCl, pH 7.5), and fractions containing the protein were combined and concentrated using an Amicon Centrifugal Filter. The UN2A protein was further purified by size exclusion chromatography (SEC) on a HiPrep 16/60 Sephacryl S-200 HR column (Cytiva) using UN2A storage buffer (10 mM HEPES and 100 mM NaCl, pH 7.5) as the mobile phase. Purity was assessed by SDS-PAGE by Coomassie staining (Fig. S1), and protein concentration was determined by the Bradford assay before storage at –80°C in 20% glycerol.

## Expression and purification of S100A1

The expression plasmid for S100A1 was transformed into BL21(DE3) cells and grown overnight in LB media + ampicillin (100 µg/ml). The overnight culture was used to inoculate a larger culture of fresh media; and the culture was grown at 37°C and 220 rpm to $OD_{600}$ ~0.7, and then induced with 1 mM IPTG and grown overnight at 18°C and 220 rpm shaking. The overnight culture was harvested, and the cell pellet was resuspended in AEX Buffer A (50 mM Tris and 10 mM EDTA, pH 7.5) with lysozyme (250 µg/ml), DNAse (40 µg/ml), $MgCl_2$ (10 mM), and Triton X-100 (1% vol/vol). The resuspended cells were put through one freeze–thaw cycle and then were further lysed with a French press at 25,000 psi, and insoluble material was removed via centrifugation at 15,000 × *g*. The resulting supernatant was loaded on a Capto-Q anion exchange column (Cytiva) with Buffer A (50 mM Tris, 10 mM EDTA, and 0.5 mM TCEP, pH 7.5) as the initial mobile phase. The protein was eluted with a 0–100% gradient using Buffer B (50 mM Tris, 1 M NaCl, 10 mM EDTA, and 0.5 mM TCEP, pH 7.5). The gradient fractions were analyzed by SDS-PAGE, pooled, and dialyzed into HIC Buffer A (10 mM Tris, 500 mM NaCl, 10 mM CaCl2, and 0.5 mM TCEP, pH 7.5). The dialyzed protein was loaded onto a GE phenyl Sepharose column and eluted with a 100% step gradient using HIC Buffer B (10 mM Tris, 500 mM NaCl, 10 mM EDTA, and 0.5 mM TCEP, pH 7.5). SDS-PAGE and Coomassie staining were used to determine the purity, and the fractions with the protein of interest were pooled (Fig. S2). The concentration of protein was determined using NanoDrop at 280 nm. The protein was stored in 20% glycerol at –80°C.

## Expression and purification of FRET clones

The expression plasmid containing the *UN2A* gene inserted between cyan fluorescent protein (CFP) and yellow fluorescent protein (YFP) was transformed into BL21(DE3) pLysS cells and grown overnight. The overnight culture was inoculated into fresh LB media + ampicillin (100 µg/ml) + chloramphenicol (34 µg/ml) and was incubated at 37°C until $OD_{600}$ ~0.6 was reached. The culture was induced with 1 mM IPTG and incubated at 18°C for 16–18 h. The overnight culture was harvested and resuspended in IMAC loading buffer (50 mM $Na_2HPO_4$, 200 mM NaCl, 25 mM Imidazole, and pH 8) with lysozyme (250 µg/ml), DNAse (40 µg/ml), $MgCl_2$ (10 mM), and Triton X-100 (1% vol/vol). The resuspended pellet was lysed using a combination of freeze–thaw, chemical lysing, and French press. Insoluble proteins were removed via centrifugation at 15,000 × *g*, and the resulting supernatant was loaded into a His-Trap column (Cytiva). UN2A-fluorescent resonance energy transfer (FRET) or 12AA protein were eluted using a 0–100% gradient of elution buffer (50 mM $Na_2HPO_4$, 200 mM NaCl, and 250 mM imidazole, pH 8). Purity was assessed by SDS-PAGE using Coomassie staining (Fig. S3), and protein concentration was determined by the Bradford assay before storage at –80°C in 20% glycerol. The control plasmid containing a 12–amino acid linker between the CFP and YFP (Ohashi et al., 2007) was expressed and purified using the same methodology.

## Co-immunoprecipitation (Co-IP) assay

Co-IP experiments were performed using the Pierce Classic Magnetic IP/Co-IP Kit (Thermo Fisher Scientific) to assess the interaction between S100A1 and UN2A in the presence of $Ca^{2+}$. 500 µg of S100A1, 500 µg of UN2A, and 5 µg of anti-S100A1 antibody (1 mg/ml) were diluted to a final volume of 500 µl in the presence of $Ca^{2+}$ (250 mM NaCl, 20 mM HEPES, and 1 mM $Ca^{2+}$, pH 6). The samples were gently rotated and incubated overnight at 4°C to allow binding.

The following day, 25 µl of the magnetic beads provided in the kit were prewashed twice with wash buffer (250 mM NaCl, 20 mM HEPES, 0.05% Tween-20, and 1 mM $CaCl_2$, pH 6). The samples were added to the prewashed beads and incubated with rotation for 1 h at 4°C to capture the immune complexes. The beads were collected with a magnetic stand, and the unbound sample was removed and saved for analysis. The beads were washed three times with 500 µl of wash buffer and once with 500 µl of ultrapure water, discarding the supernatant after each wash.

100 µl of the low-pH elution buffer provided in the kit was added to the beads, and the samples were incubated at room

temperature for 10 min with gentle mixing. The beads were collected with the magnetic stand, and the supernatant containing the eluted protein complexes was saved. To neutralize the low pH, 10 μl of neutralization buffer was added to each 100 μl of eluate. The eluted samples were analyzed by SDS-PAGE, followed by silver staining to detect S100A1 and UN2A. Densitometry calculations were performed using ImageJ to quantify band intensities in the Co-IP assays.

### HPLC SEC
Purified UN2A-FRET and S100A1 were diluted in the absence of $Ca^{2+}$ (150 mM NaCl, 20 mM HEPES, and 1 mM EGTA, pH 7.4) or in the presence of $Ca^{2+}$ (150 mM NaCl, 20 mM HEPES, and 1 mM $CaCl_2$, pH 7.4). Samples were prepared by mixing UN2A-FRET at a concentration of 20 μM and S100A1 at a concentration range from 0 to 25 μM and were incubated for 30 min at room temperature. Each sample was separated on a TSKgel G2000SWxl HPLC column at 0.700 ml/min using an Agilent Technologies 1260 Infinity LC system. The separation range for the TSKgel G2000SWxl HPLC column is 5,000–150,000 Da. Peaks were detected at 280 nm using a Gilson 170 Diode Array detector, and the retention time was determined using the time when maximum absorbance occurred. The resulting chromatograms were collected and integrated using the Agilent OpenLabCDS software. The standard deviation for each set of HPLC measurements was calculated to ensure the precision of the results.

### Circular dichroism
Circular dichroism (CD) experiments were performed using a JASCO J-1500 CD Spectrometer at 25°C using a Xe lamp and a 1-mm quartz cuvette. The data pitch was 0.2 nm, using a continuous scanning mode of 100 nm/min and a bandwidth of 2.00 nm. The resulting spectra were an average of three accumulations and were collected between 190 and 250 nm. Purified UN2A samples were prepared at 10 μM in 20 mM Tris, 10 mM $KH_2PO_4$, and 1 mM EGTA at a pH range of 4–8. The data between 190 and 200 nm were discarded due to buffer noise, and the data from 200 to 250 nm are shown in the Fig. 5 A.

### FRET measurements
Samples were prepared by mixing purified protein at a concentration of 1 μM in either saturating $Ca^{2+}$ buffer (10 mM HEPES, 100 mM NaCl, and 1 mM $CaCl_2$) or $Ca^{2+}$-free buffer (10 mM HEPES, 100 mM NaCl, and 1 mM EGTA) at pH 6 and pH 7.4. Samples were incubated for 1 h to ensure that the binding had reached equilibrium, and then the emission spectra were measured using a JASCO FP-8500 fluorescence spectrometer at room temperature. Each sample was excited at 433 nm and the emission was measured from 445 to 600 nm at 0.5-nm intervals using 1-nm excitation slits and 5-nm emission slits.

### Surface plasmon resonance (SPR) measurements
SPR experiments were performed with a two-channel OpenSPR instrument (Nicoya Lifesciences) at 25°C with a 100-μl sample loading loop. The carboxyl sensor surface was conditioned with 10 mM HCl at a flow rate of 150 μl/min and was then functionalized using an EDC/NHS solution from the OpenSPR

Carboxyl Reagent Kit at a flow rate of 20 μl/min. UN2A was immobilized on the active channel of a Nicoya High-Capacity Carboxyl Sensor (Nicoya Lifesciences) at a flow rate of 20 μl/min. Ligand immobilization was 66 response units for these experiments. Ethanolamine-blocking solution was then used to deactivate the remaining active sites on the carboxyl sensor at a flow rate of 20 μl/min. Binding of S100A1 was tested in the absence of $Ca^{2+}$ (250 mM NaCl, 20 mM HEPES, 0.05% Tween, and 1 mM EGTA) and in the presence of $Ca^{2+}$ (250 mM NaCl, 20 mM HEPES, 0.05% Tween, and varying $CaCl_2$ concentrations to reach pCa 7, 5, and 3) at pH 6 and pH 7.4. S100A1 was injected in concentrations from 5 to 100 μM for 2 min to allow association and was allowed to dissociate for 5 min prior to regeneration. Following each S100A1 injection, the regeneration buffer (250 mM NaCl, 20 mM HEPES, and 1 mM EDTA) was injected to completely dissociate all S100A1 bound and regenerate the surface. The corrected data were imported into TraceDrawer Analysis Software and were used to calculate kinetic parameters, fitting the data to a 1:1 model. The standard deviation for each set of measurements was calculated to ensure the precision of the results.

### Online supplemental material
Supplementary information includes amino acid sequences for UN2A, S100A1, UN2A-FRET, gel images of purified proteins (Figs. S1, S2, and S3), SEC chromatograms of UN2A with and without $Ca^{2+}$ (Fig. S4), nonspecific binding data for S100A1 to the sensor chip (Fig. S5), SPR affinity fits (Figs. S6, S7, S8, S9, S10, and S11), and sequence alignment of RYR and UN2A (Data S1).

## Results
### S100A1 binds to the UN2A region in titin
Recent studies have highlighted the importance of the N2A region as a signaling hub within the extensible region of titin, with interactions to the MARP family of proteins, Calpain3, and SMYD2 with a chaperone of methylated Hsp90 all being localized to this area (Miller et al., 2003; Sorimachi et al., 1995; Hayashi et al., 2008; Donlin et al., 2012; Voelkel et al., 2013). Work in our lab has demonstrated that $Ca^{2+}$ modulates the interaction between N2A and actin, but the mechanism of that interaction was initially unclear (Zhou et al., 2021; Dutta et al., 2018; Tsiros et al., 2022). One potential hypothesis was that the observed $Ca^{2+}$ dependence could be due to interactions of $Ca^{2+}$-binding proteins with this region. We therefore performed a sequence analysis of the N2A region to determine if there might be an interaction between $Ca^{2+}$-binding proteins known to exist in the sarcomere and titin's N2A region.

One of the more ubiquitous $Ca^{2+}$-binding regulatory proteins in the muscle is S100A1, which has been linked to a number of roles in muscle function, including regulation of the RYR and binding to the PEVK region of titin (Sun and Kekenes-Huskey, 2020; Yamasaki et al., 2001). We analyzed the sequence of the UN2A region and found a potential binding site with weak homology to the consensus S100A1-binding sequence (K/R)(L/I) XWXX(I/L)L existing in the helical region of UN2A (Ivanenkov

et al., 1995; Prosser et al., 2008). The binding sites for S100 proteins commonly occur on alpha helical structures (Prosser et al., 2008), so the putative binding site is consistent with other known S100 interactions. Interestingly, we performed a sequence alignment with RYR and found that there is not much similarity between the UN2A region and the S100A1-binding sites on the RYR (Data S1). However, given lack of a specific consensus binding sequence, we hypothesized that S100A1 might bind to the UN2A region.

To test the hypothesis that S100A1 interacts with the UN2A region of titin, we conducted both SEC experiments and Co-IP. We expressed and purified S100A1 and a previously created construct, where the UN2A sequence inserted between CFP and YFP UN2A (Tiffany et al., 2017) and looked for binding using SEC. Initial tests with the purified UN2A and S100A1 demonstrated that the two proteins exhibited nearly identical retention times, which would have complicated the analysis of any binding reaction. We therefore used the UN2A-FRET clone so that the two individual proteins had unique retention times, making it clearer if binding is occurring. Based on our model, we predicted that if there was an interaction between UN2A and S100A1, the observed peak would shift to shorter retention times based on the increased size of the complex. The purified protein was mixed at equimolar ratios and separated on TSKgel G2000SWxl HPLC column at varying $Ca^{2+}$ concentrations. At low $Ca^{2+}$ concentrations (pCa 10), there was no shift in the retention time of the UN2A-FRET protein, suggesting that there was no binding (Fig. S4). The same experiment was conducted at pCa 3 and there was a reproducible shift of the UN2A-FRET peak to faster retention times, consistent with formation of a complex between the S100A1 and the UN2A-FRET clone (Fig. 2 A). The magnitude of the shift was small, but this was expected as the UN2A-FRET clone is three times the size of the S100A1 and is already in an extended state, so binding of S100A1 will not result in a large shift in the Stokes radius. We further tested the nature of the binding equilibrium by varying the concentration of S100A1 and determined that the retention time shift increased with increasing S100A1 concentration in the presence of $Ca^{2+}$, consistent with the equilibrium between the bound and unbound states shifting toward the bound state as a function of S100A1 concentration (Fig. 2, A and C). In contrast, there was no shift in retention time with increasing S100A1 concentration in the absence of $Ca^{2+}$ (Fig. 2, B and C). These results support a model in which S100A1 binds to the N2A region of titin through a concentration-dependent equilibrium that is regulated by $Ca^{2+}$.

Co-IP was used to further validate the interaction between S100A1 and UN2A. For these experiments, 500 µg of S100A1 and 500 µg of UN2A were incubated together, which provided an ~1.3:1 M ratio of S100A1 to UN2A. Both proteins were incubated together, and then the S100A1 was captured using an anti-S100A1 antibody and Protein-G labeled magnetic beads. The UN2A coprecipitated with the captured S100A1, but was not observed in control samples that did not contain S100A1 (Fig. 3), consistent with our HPLC results. Densitometry analysis using ImageJ showed that the band intensity of the N2A was 25% of the intensity of the S100A1 band, suggesting a binding ratio of ~1:4

for UN2A to S100A1 in the complex. The relatively low-binding yield supports a weak interaction between UN2A and S100A1. While both the HPLC and Co-IP results are consistent with a binding interaction between UN2A and S100A1, they also suggest that the binding affinity is on the weaker (µM) side due to the small SEC shifts and the amount of UN2A captured in the Co-IP.

## S100A1 binds UN2A with micromolar affinity

While our SEC experiments provided evidence that S100A1 is capable of binding to UN2A, we wanted to understand the thermodynamics and kinetics of this binding more completely. To do this, we performed a series of SRP experiments to further characterize the interaction between S100A1 and UN2A. UN2A was successfully attached to the carboxyl sensor surface, whereas S100A1 could not be successfully attached to the surface. S100A1 has commonly been used as the analyte in other SPR experiments rather than the ligand, suggesting that there may be an issue attaching S100A1 to SPR surfaces (Leclerc et al., 2009; Grycova et al., 2015; Van Dieck et al., 2010). S100A1 was titrated over the surface at varying concentrations to determine the binding thermodynamics of the system. Initial experiments were conducted at pH 7.4 and pCa 3 to ensure conditions that would theoretically produce the tightest binding. Binding was measured at a range of S100A1 concentrations, and the $K_D$ was determined to be 21.6 µM ([Fig. 4, A–C], nonspecific binding is shown in Fig. S5 and affinity fits shown in Figs. S6, S7, S8, S9, S10, and S11). Interestingly, the dissociation phase of the binding curve appears to have two profile steps, a faster dissociation phase and a slower dissociation phase. Despite this observation, we opted to use a single-state model to fit our data since there is no experimental evidence to support a two-state dissociation, and the single-state verse two-state dissociation did not provide significant improvements to the fits. The determined dissociation constant is on the weaker side of dissociation constants for protein–protein interactions. However, if this interaction serves a regulatory purpose, it is possible that if this interaction was tight that it would be unfavorable, as will be discussed in more detail later.

Our SEC experiments demonstrated that S100A1/UN2A binding was $Ca^{2+}$ dependent, so we repeated the SPR experiments at three additional $Ca^{2+}$ concentrations: pCa 5, pCa 7, and pCa 10. This range covers both the concentration of $Ca^{2+}$ in the resting muscle (pCa 10) to concentrations over which muscle becomes activated and the range where the binding sites on the S100A1 become saturated by $Ca^{2+}$. No significant binding was observed at either pCa 10 or 7, while increasing binding was observed at both pCa 5 and 3 (Fig. 4 B). The binding affinity at pCa 5 was calculated to be 285 µM, which is 10-fold weaker than the binding at pCa 3. S100A1 has two $Ca^{2+}$-binding sites, with binding affinities of 10–50 µM (pCa 5.0–4.3) and 200–500 µM (pCa 3.7–3.3). This suggests that binding is $Ca^{2+}$ dependent since at least one of the binding sites would become occupied around pCa 5 and the second binding site would become occupied around pCa 3, which provides a potential model for why binding significantly increases between pCa 5 and pCa 3.

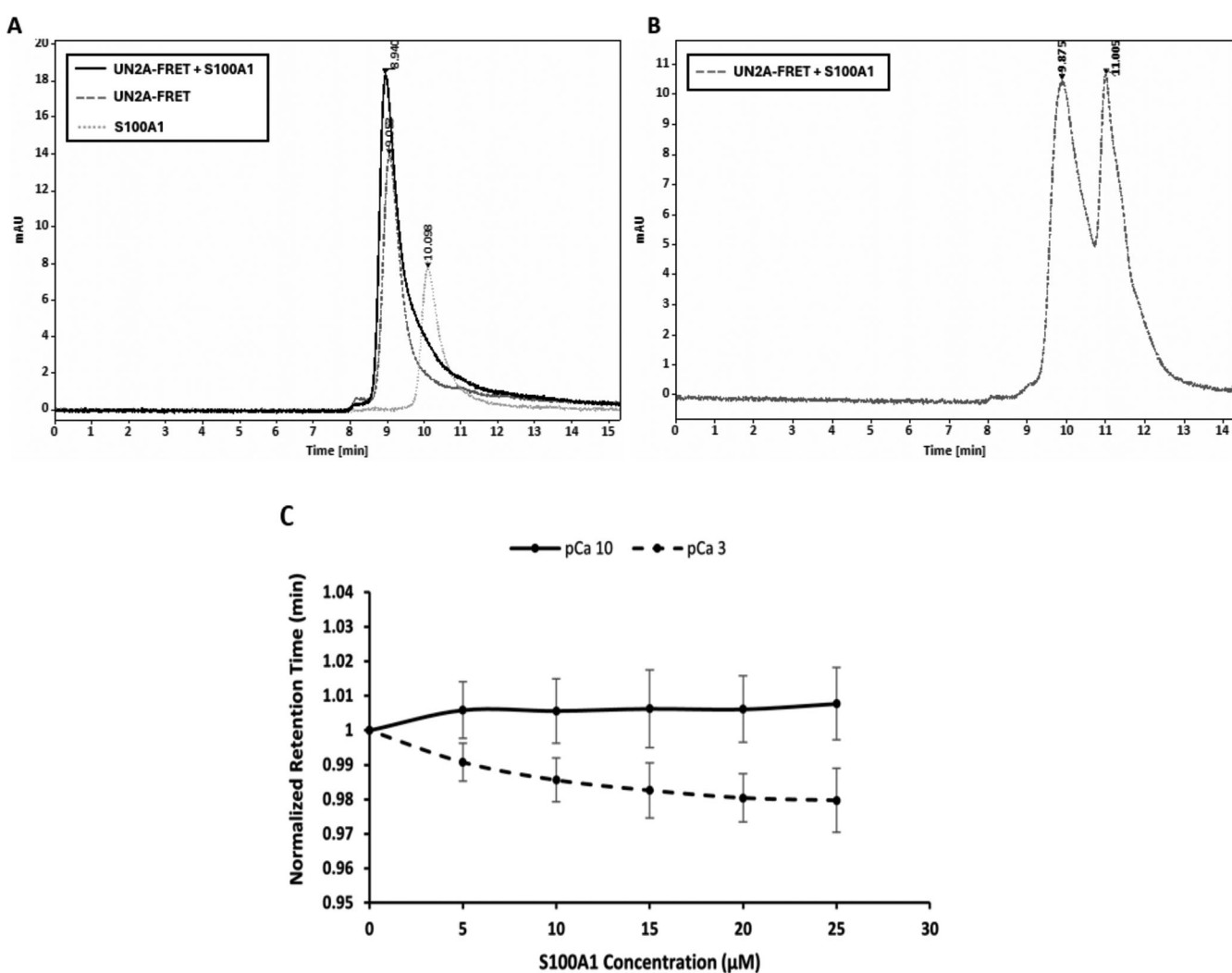

Figure 2. **HPLC binding data for UN2A and S100A1 in the presence and absence of Ca²⁺. (A)** S100A1 binds UN2A in the presence of Ca²⁺. Purified UN2A (dark gray, dashed line) and S100A1 (light gray, dotted line) were incubated at pCa 3 and separated on TSKgel G2000SWxl HPLC column. The retention time of the UN2A/S100A1 complex (black, solid line) decreases with increasing Ca²⁺ concentration, suggesting a Ca²⁺-dependent binding interaction. **(B)** S100A1 does not bind UN2A in the absence of Ca²⁺. Purified UN2A and S100A1 were incubated at pCa 10 and separated on TSKgel G2000SWxl HPLC column. In the absence of Ca²⁺, the UN2A/S100A1 complex (dashed line) shows two separate peaks corresponding to UN2A (retention time = 9.875 min) and S100A1 (retention time = 11.005 min), indicating no binding interaction. **(C)** Effect of Ca²⁺ and S100A1 concentration on UN2A's retention time. Plots depict the normalized retention time of UN2A at 20 µM relative to varying concentrations of S100A1 at pCa 3 (dashed line) and pCa 10 (solid line). The retention times were normalized to mitigate experimental variations. At pCa 3, a decrease in the retention time of UN2A suggests a complex formation at higher S100A1 concentrations. At pCa 10, the retention time of UN2A remains stable, indicating no complex formation with S100A1.

### Binding of S100A1 induces conformational change in UN2A

It is well established that structure can be induced in disordered regions when bound interacting with their binding partner. We therefore hypothesized that a conformational change might occur in the UN2A when S100A1 binds. To test this hypothesis, we utilized the FRET reporter used in our HPLC experiments (Tiffany et al., 2017). This reporter system contains the UN2A region between CFP and YFP proteins, providing a way to measure conformational changes that occur on the UN2A sequence. Increased YFP signal at 527 nm should occur if the UN2A shifts to a more compact structure, while YFP fluorescence will decrease at 527 nm if the UN2A assumes a more elongate conformation. This provided a tool to determine if S100A1 binding to UN2A induces conformational changes.

Fluorescence emission spectra of UN2A and UN2A/S100A1 in the presence and absence of Ca²⁺ at pH 7.4 were collected by exciting CFP at 433 nm and the emission was measured from 445 to 600 nm. No significant changes were observed for UN2A/S100A1 (Fig. 5 A) compared with UN2A in the absence of Ca²⁺, as would be expected if there was no interaction between the two proteins. When the experiment was repeated at pCa 3, there was decreased YFP fluorescence in the UN2A reporter (Fig. 5 B), suggesting that S100A1 binding induces a conformational shift that increases the distance between the two fluorophores. This shift indicates that UN2A is forming a more extended conformation. FRET was used to calculate an estimate of the end-to-end distance of UN2A as described in Tiffany et al. (2017). Based on each trial, S100A1 binding induced a shift of 6–8 Å in the end-to-end distance of UN2A.

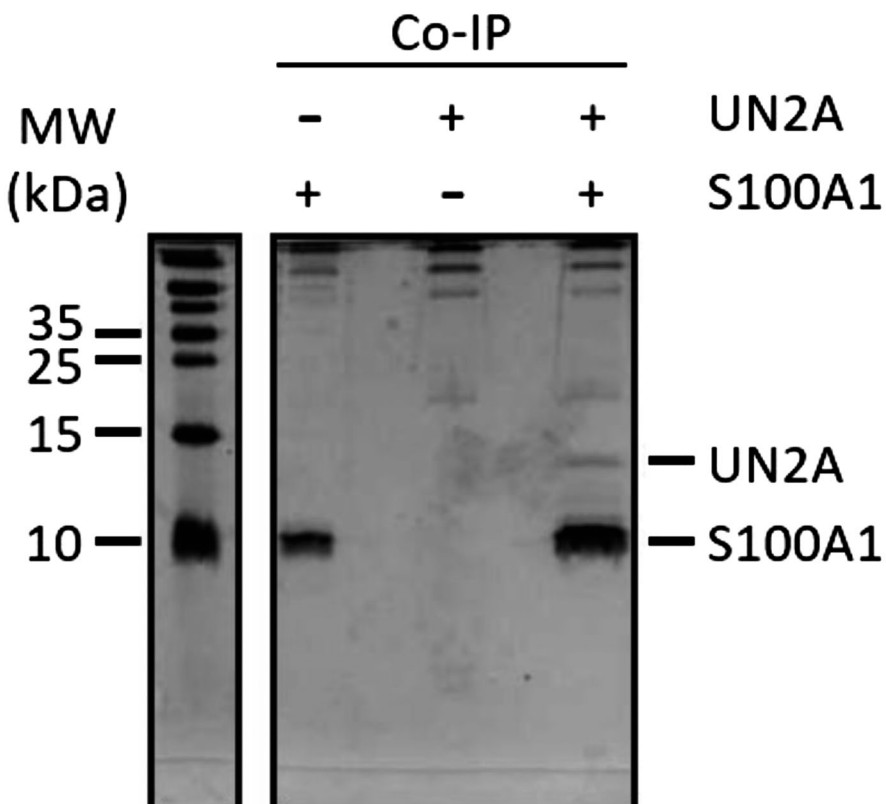

Figure 3. **Co-IP shows capture of UN2A by S100A1.** Purified UN2A and S100A1 were incubated together, and the S100A1 was precipitated using an α-S100A1 antibody. Samples were separated by SDS-PAGE and analyzed by silver-staining. Lanes 1 and 3 are controls of the individual proteins showing that the antibody captures the S100A1 but not the UN2A. The presence of the UN2A along with the S100A1 in lane 2 supports our hypothesis that the S100A1 and UN2A interact with S100A1 is in the Ca²⁺-bound state. The contrast and brightness of the original image was adjusted slightly to improve the signal to noise between the gel and the bands. Source data are available for this figure: SourceData F3.

## UN2A undergoes a pH-dependent conformational change

The UN2A region is a unique structural element in titin, containing a tri-helix bundle with disordered regions on either side, while the majority of titin's structure consists of β-sheet structure. In analyzing the sequence of the disordered regions in the UN2A region, similarities to the poly-E regions in the disordered PEVK region were observed. Previous work from our lab has demonstrated that these sequences can undergo pH-triggered conformational changes, so we hypothesized that this region of titin might also undergo similar shifts (Sudarshi Premawardhana et al., 2020). To test this possibility, we collected far-UV CD spectra for the UN2A region at a range of pHs and observed increased helical structure as pH decreases (Fig. 6, A and B). We predicted that we would observe a sigmoidal curve, similar to what was observed with our experiments on the poly-E region of titin; however the observed transition appears to be more linear. In hindsight, the lower number of glutamate clusters in UN2A relative to the PEVK constructs we have tested would reduce the overall charge repulsion, so the lack of a significant sigmoidal transition is perhaps not surprising.

## S100A1/UN2A binding increases with decreased pH

S100A1 tends to bind to helical regions in their target proteins, so we hypothesized that S100A1 might bind more tightly at lower pH due to the increased helical structure observed for UN2A. We therefore repeated the SPR binding experiments at pH 6.0, predicting that binding affinity will increase as pH decreases and there is more helical structure in the UN2A protein. As can be seen in the Fig. 7 A, the measured binding more than

doubled at pH 6.0, consistent with our hypothesis. We repeated this experiment at pH 6.5, predicting that the binding would be between what was observed at pH 7.4 and what was observed at pH 6.0. This is exactly what was observed (Fig. 8). The binding affinity increased to 9.95 µM at pH 6.5 and to 5.0 µM at pH 6.0, demonstrating that binding is stronger at lower pHs.

We repeated the SPR experiments at three additional Ca²⁺ concentrations: pCa 5, pCa 7, and pCa 10 at pH 6. No significant binding was observed at either pCa 10 or 7, while increased binding affinity was observed at both pCa 5 and 3 (Fig. 7 B). The binding affinity at pCa 5 was calculated to be 174 µM, which is 30-fold weaker than the binding at pCa 3. This suggests that binding is also Ca²⁺ dependent at pH 6, but with a stronger affinity, following the same Ca²⁺-dependence trend as seen in pH 7.4 due to the occupation of the Ca²⁺-binding sites on S100A1 around pCa 5 and pCa 3.

Having seen that binding increased at lower pH, we repeated the FRET experiments at pH 6.0 to understand how the shift in pH impacts the structure of UN2A (Fig. 9, A and B). We initially compared the signal from the UN2A in the absence of S100A1 to see if the change in helical conformation impacted the distance between the fluorescence proteins. When the two spectra are overlaid, it is clear that there is a lower FRET signal at pH 6.0 (Fig. 9 C), suggesting that the increased helical signal observed by CD is due to a lengthening of the UN2A. We then repeated the binding experiments with S100A1 present and observed a decreased FRET signal at lower pH, further suggesting that the structure induced by S100A1 binding can also be induced by decreasing the pH.

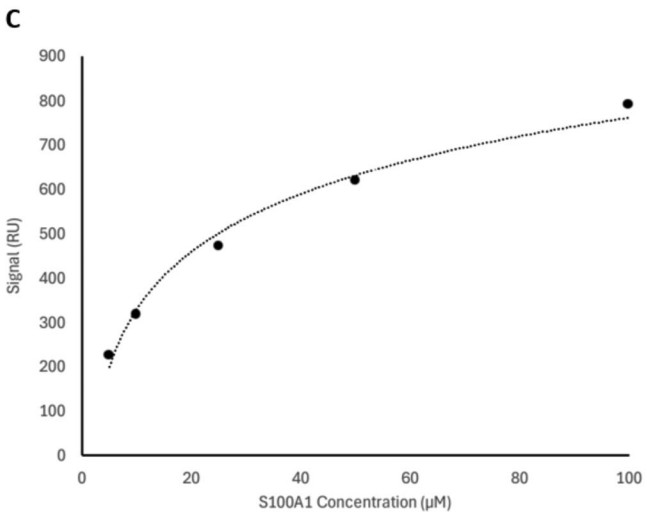

Figure 4.   **SPR binding data for UN2A with S100A1 at pH 7.4. (A)** S100A1 demonstrates concentration dependent binding by SPR. UN2A was attached to the gold surface and varying concentrations of S100A1 was titrated across the surface at pH 7.4 and pCa 3. Data were fit to a single-state dissociation model and results in a $K_D$ of 21.6 μM. **(B)** Binding affinity of S100A1 to UN2A depends on $Ca^{2+}$ concentration. SPR-binding experiments were repeated at a range of $Ca^{2+}$ concentrations, from pCa 10 to pCa 3. Negligible binding was observed at pCa 10 and 7 with increased binding at pCa 5, and the strongest binding observed at pCa 3. This is consistent with a model where weak binding occurs at pCa 5 due to one $Ca^{2+}$ bound per monomer and strong binding occurs at pCa 3 when both $Ca^{2+}$-binding sites are occupied. **(C)** Representative affinity fit for the binding interaction between S100A1 and UN2A at pCa 3.

## Discussion

$Ca^{2+}$ is a ubiquitous signaling molecule that plays a critical role in muscle activation through interactions with troponin (Gomes et al., 2002). In addition to directly acting to induce conformational changes, $Ca^{2+}$ can also act as a signaling molecule through $Ca^{2+}$-binding proteins such as S100A1 to regulate various processes such as function of the RYR. $Ca^{2+}$ regulation of titin function is a critical part of normal muscle function, though it is not as widely appreciated as it is for proteins like troponin. Interaction between $Ca^{2+}$ and titin has been shown to regulate binding of the N2A region to actin (Dutta et al., 2018; Zhou et al., 2021; Van Der Pijl et al., 2021) and modulate the stability of several Ig domains (Kelly et al., 2020, 2021; Kelly and Gage, 2021). Binding of the PEVK region to actin has been shown to be disrupted by S100A1 (Gutierrez-Cruz et al., 2001; Fukushima et al., 2010; Yamasaki et al., 2001; Labeit et al., 2003), and there

is a suggested binding site for S100A1 in the C-terminus of titin's kinase domain in the presence of $Ca^{2+}$ (Yamasaki et al., 2001; Amodeo et al., 2001). Taken together, this highlights the importance of $Ca^{2+}$ in regulating titin function.

The results presented in this manuscript highlight a previously undiscovered interaction between the N2A region of titin and S100A1. An analysis of the N2A region identified a potential binding site for S100A1 in the UN2A region, which was confirmed by SEC. SPR determined that the equilibrium constant for this interaction at pH 7.4 and pCa 3 was 21.6 μM, though it increases with decreasing $Ca^{2+}$ concentrations. This suggests that binding is regulated in part through intracellular $Ca^{2+}$ concentrations. Our results also demonstrated that binding induces a conformational change in UN2A. Most interestingly, isolated UN2A protein gains helical conformation as pH decreases and binding affinity increases at lower pHs as well.

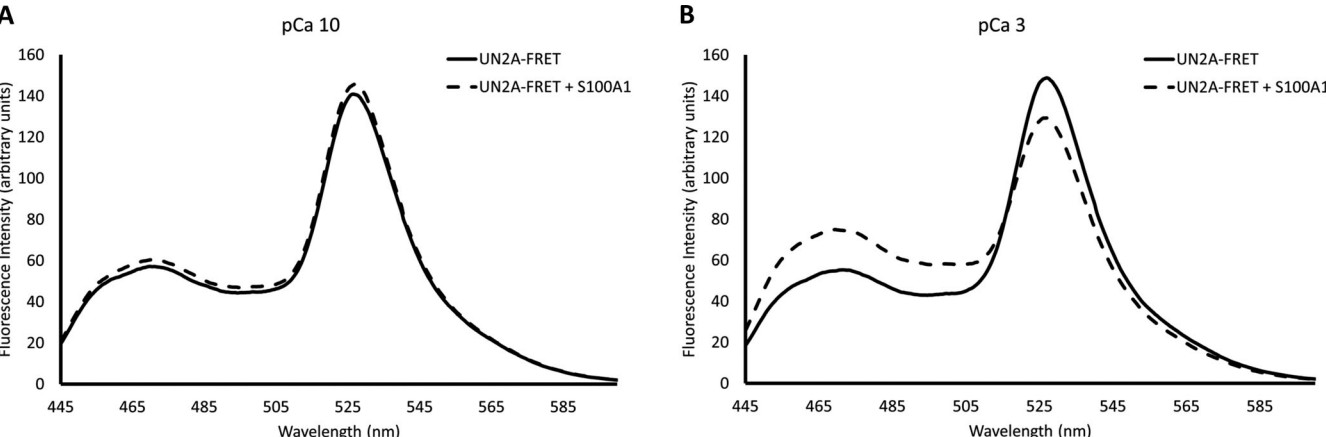

**Figure 5. UN2A undergoes a conformational change with S100A1 binding.** The UN2A region was cloned in between CFP and YFP to create a FRET reporter system to observe conformational changes in UN2A. Samples were excited at 433 nm and fluorescence emission spectra were measured from 445 to 600 nm. **(A and B)** Binding experiments were conducted in both low $Ca^{2+}$ (pCa 10, A) and high $Ca^{2+}$ (pCa 3, B) conditions. No significant differences were observed in low $Ca^{2+}$ conditions. Under high $Ca^{2+}$ conditions, there was an increase in the CFP (465 nm) peak and a decrease in the YFP (525 nm) peak, consistent with lower FRET due to an increase in the distance between the two fluorophores. This suggests that the UN2A adopts a more elongated conformation when S100A1 binds.

Taken together, these results provide evidence that titin/S100A1 binding is regulated by both $Ca^{2+}$ and pH, and we propose that the role of this interaction may be to regulate titin function in response to environmental changes in the cell.

### Mechanism of binding

Based on the evidence that both $Ca^{2+}$ and pH regulate S100A1–titin binding, we hypothesize that this interaction is important for tuning titin function based on the environment in the sarcomere. Binding under physiological pH (7.2–7.4) is driven by the $Ca^{2+}$ concentration, with binding beginning between pCa 7 and pCa 5, where muscle begins to be activated (Fabiato and Fabiato, 1978; Metzger and Moss, 1990) and increases as pCa

increases. This means that binding will increase the longer the muscle is repeatedly activated since $Ca^{2+}$ levels rise as the muscle remains in an activated state. Initial binding is relatively weak (∼280 μM) when the muscle initially becomes activated. However, the binding constant increases 10-fold between pCa 5 and pCa 3–21.6 μM. This is still on the weaker side of protein–protein interactions but, as will be discussed later, this may be important in relation to the potential role of this interaction.

One of the other physiological responses to repeated or prolonged activation of a muscle is decreased pH from formation of lactic acid or $CO_2$, depending on how the muscle is producing energy. Our data demonstrate that binding also becomes stronger as the pH decreases, though not as significantly as the

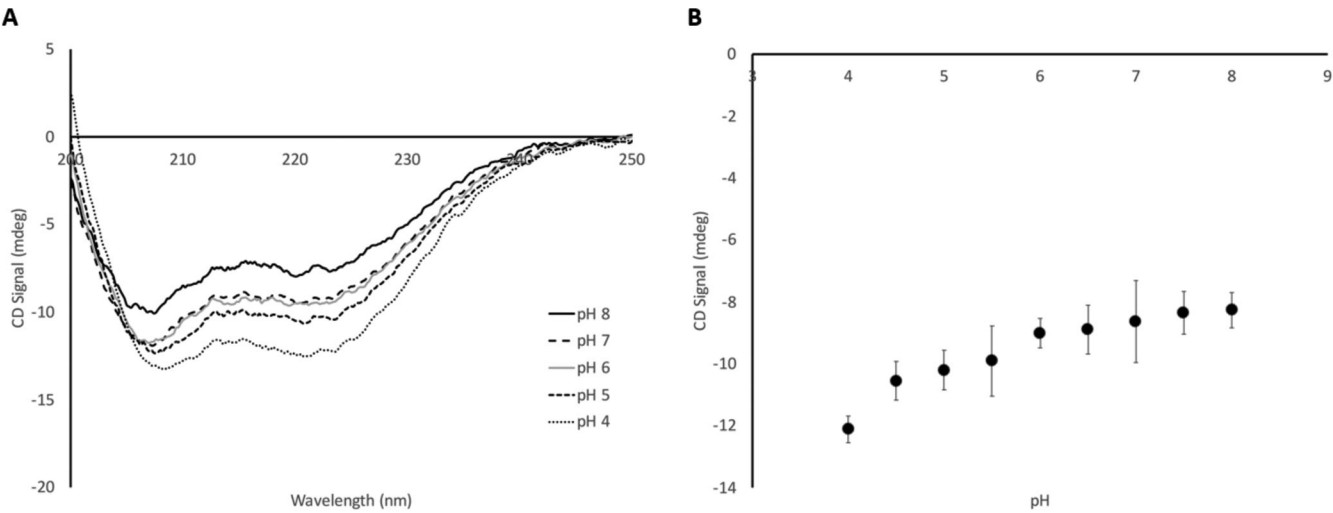

**Figure 6. UN2A undergoes a pH-dependent conformational change.** CD was used to measure the secondary structure of UN2A at a pH range from 4.0 to 8.0. **(A)** Representative CD spectra of UN2A over a range of pHs from 200 to 250 nm. As can be seen, the minima at 208 and 222 nm become deeper as the pH decreases, consistent with helical content at lower pHs. **(B)** The average mdeg signal at 222 nm was plotted as a function of pH for N = 4 trials. There is a slight decrease in the average signal between pH 8 and 6 with a more significant shift occurring below pH 6.

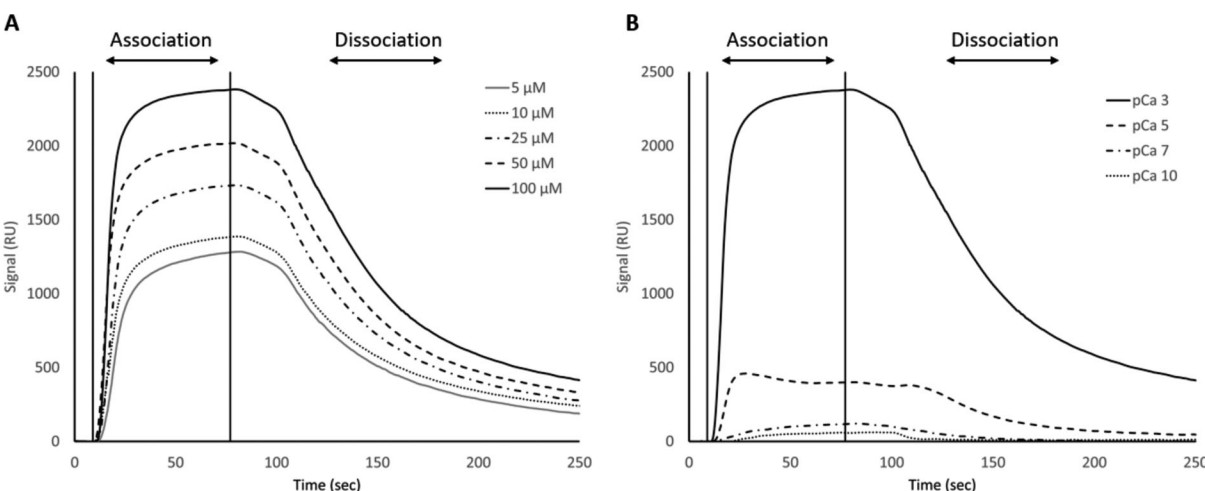

Figure 7. **SPR binding data for UN2A with S100A1 at pH 6.0. (A)** Binding affinity of S100A1 increases at pH 6.0. SPR experiments were repeated at pH 6.0 at pCa 3.0. The signal for 100 µM S100A1 increases from ~800 RU at pH 7.4 to ~2,400 RU at pH 6.0. The calculated binding affinity ($K_D$) increased from 21.6 µM at pH 7.4 to 5.0 µM at pH 6.0. **(B)** $Ca^{2+}$ dependence of S100A1 binding occurs at lower pH similar to pH 7.4. SPR-binding experiments were repeated at a range of $Ca^{2+}$ concentrations, from pCa 10 to pCa 3 at pH 6.0. Similar to at pH 7.4, negligible binding was observed at pCa 10 and 7 with increased binding at pCa 5, and the strongest binding observed at pCa 3. RU, response units.

shift observed with changes in $Ca^{2+}$ concentration. Overall, however, our work supports a model where binding of S100A1 to titin is dependent on two variables, allowing the equilibrium between bound and unbound S100A1 to shift in response to these variables.

The mechanism of how these two variables modulate binding appears to be dependent on the variable. S100A1 has two $Ca^{2+}$-binding sites, which have binding constants of 200–500 µM at the N-terminal region and 10–50 µM at the C-terminal region, which correlates with pCa 3.7–3.3 and pCa 5.0–4.3, respectively.

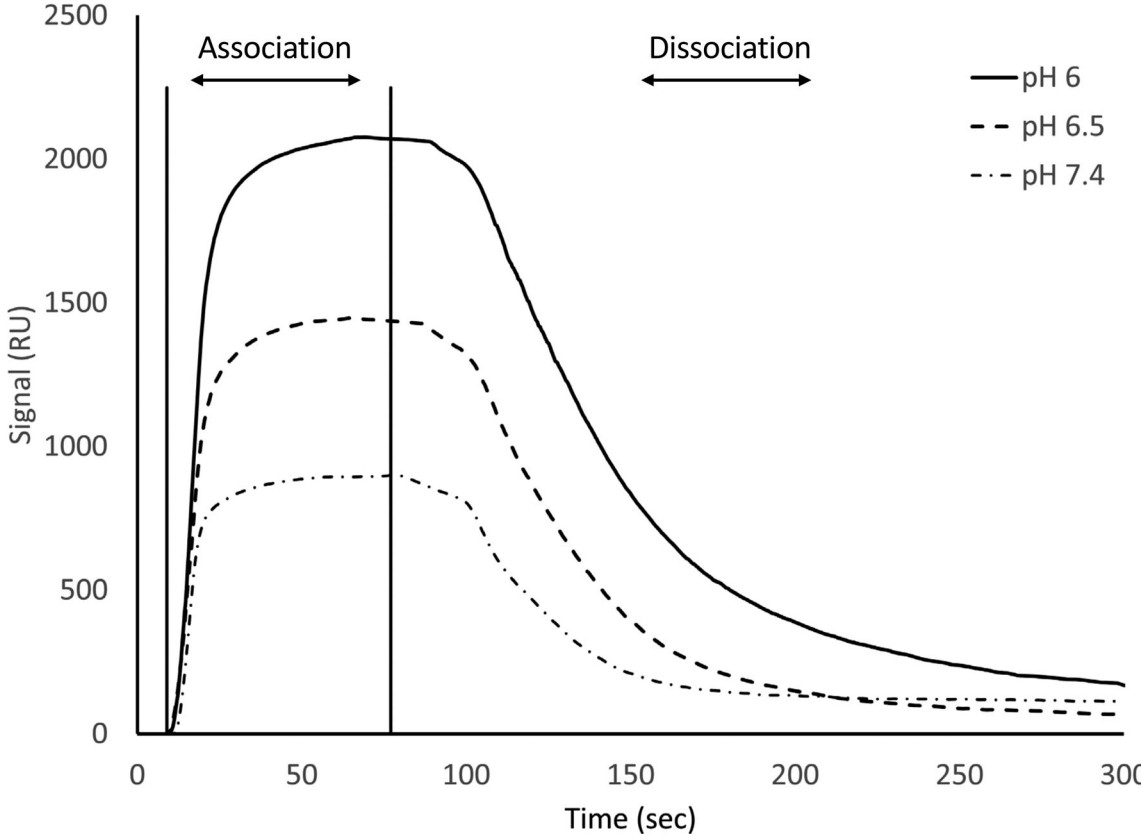

Figure 8. **Binding affinity of S100A1 demonstrates a dependence on pH.** SPR experiments were repeated at pH 6.0, pH 6.5, and pH 7.4 at pCa 3.0 and 100 µM S100A1. The max point on the sensorgram increased as a function of decreasing pH, suggesting a pH dependence to the binding of S100A1 to UN2A.

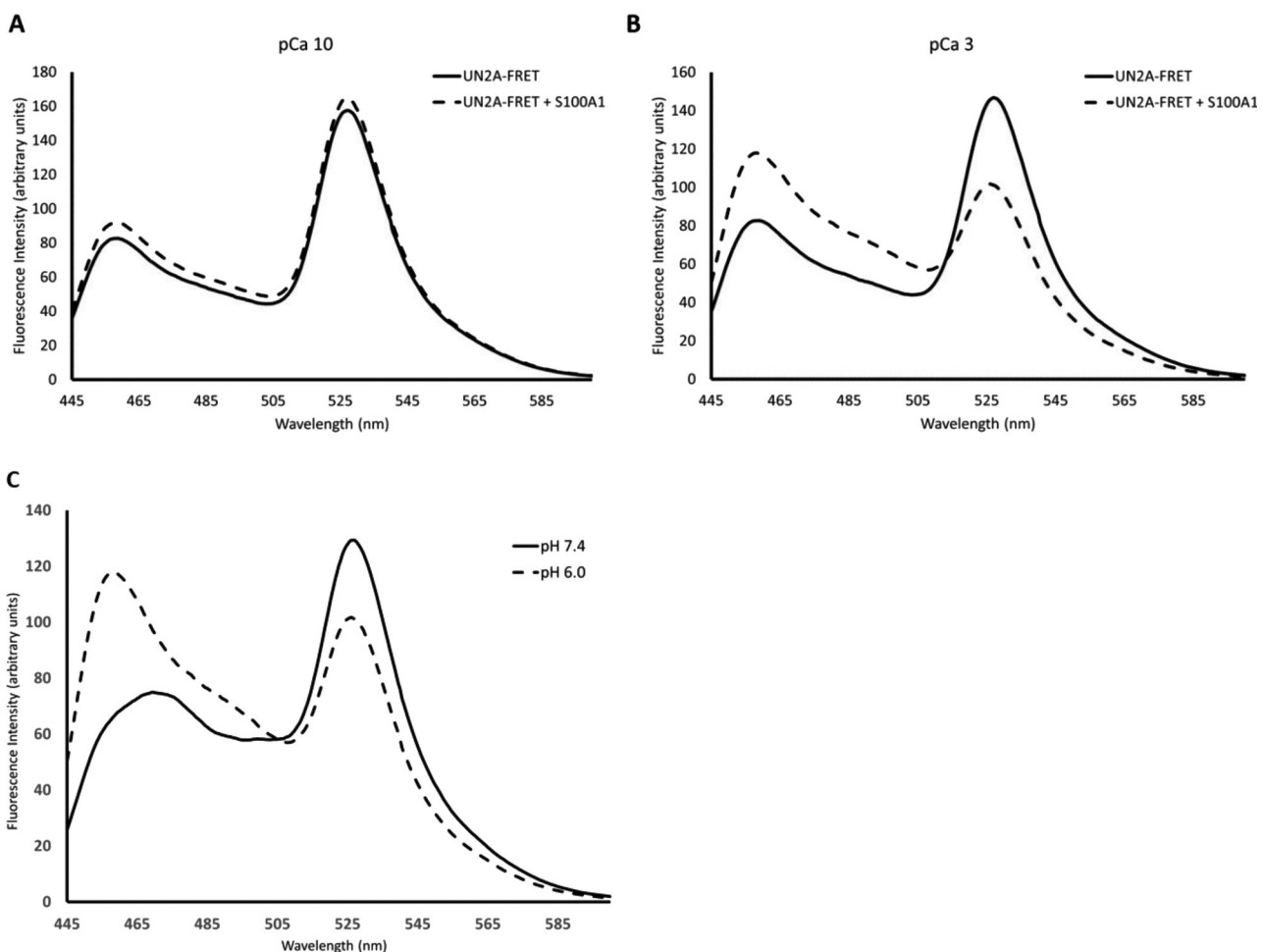

**Figure 9. Larger conformational change in UN2A occurs at pH 6.0 with S100A1 binding.** FRET experiments were performed at pH 6.0 using the same conditions as the initial experiments at pH 7.4. **(A)** Similar to pH 7.4, there was no significant differences were observed in low $Ca^{2+}$ conditions. **(B)** Under high $Ca^{2+}$ conditions, there was an increase in the CFP (465 nm) peak and a decrease in the YFP (527 nm) peak, though the magnitude of the change is greater than was observed at pH 7.4, suggesting that there is a greater conformational shift with S100A1 binding at pH 6.0. **(C)** Fluorescence spectra of UN2A-FRET clone in presence of S100A1 at pCa 3 at pH 7.4 and pH 6.0 overlaid to highlight the shift in the FRET signal at lower pH. There is a larger CFP (465 nm) peak and smaller YFP (527 nm) peak at pH 6.0, consistent with decreased FRET.

This is consistent with the shifts in binding we observed in our experiments, with the initial weak binding correlating with the initial $Ca^{2+}$-binding event on S100A1. The significant shift in binding occurs between pCa 5 and 3, which is where all the $Ca^{2+}$-binding sites are saturated on S100A1. The dimeric state of S100A1 could also potentially regulate this binding further, if binding occurred in both the monomeric and dimer states, though this is purely speculative currently.

Regulation of the UN2A/S100A1 interaction by pH appears to occur through the UN2A protein. Our results suggest that there is increased helical content in the UN2A protein as pH decreases, which we predict is due to the high glutamic acid content in the region of the tri-helix core. We hypothesize that this increased helical content facilitates tighter binding between UN2A and S100A1. This is supported by FRET experiments, demonstrating that there is a conformational shift in UN2A when S100A1 binds. We propose that decreasing pH mimics the effects of S100A1 binding on UN2A, resulting in a conformation that promotes stronger S100A1 binding.

**Significance of weaker binding**

The binding between S100A1 and UN2A is on the weaker end of the protein–protein interaction spectrum and would seem unimportant at initial glance. However, our model of function would suggest that a weak interaction is actually beneficial for regulating muscle function. Our current work has not identified a mechanism for this interaction. However, S100A1 has previously been shown to bind to the PEVK region of titin and to disrupt interactions between titin and actin, though the functional significance of this interaction has still not been established (Fukushima et al., 2010; Yamasaki et al., 2001). Recent work from our lab has shown that the N2A region of titin also binds actin, making it a reasonable hypothesis that the interaction between S100A1 and UN2A might play a similar function as with the PEVK region and regulate binding between N2A and actin.

The data presented in this manuscript further support this proposed model. One of the current models for titin's role in the active muscle is that there is an interaction between the N2A

region and actin, which reduces the extensible region of titin. If this interaction is critical for function, then it would not be very favorable to have another interaction that disrupted this binding. However, the longer a muscle remains active, the more potential exists for fatigue or damage. It would therefore be advantageous to have an interaction that disrupted N2A/actin binding, relaxing the strain on the muscle and reducing potential damage. The binding of S100A1 to UN2A could play that function in the muscle, consistent with the weak overall binding so that not all titin/actin interactions are disrupted.

Further supporting this hypothesis is how the affinity of S100A1 for UN2A increases with changes in pH and $Ca^{2+}$ concentration. There would be little need to disrupt the interaction between titin and actin when the muscle is in a resting state, so the lack of binding at pCa 10 makes sense. However, the longer the muscle is activated, the higher the $Ca^{2+}$ concentration will become and the more active a muscle will become. Therefore, there is a greater potential for damage, so increased S100A1 binding with increasing $Ca^{2+}$ would help reduce some of this stress on the muscle. Similarly, the longer a muscle stays active, the more metabolism that it undergoes and therefore the lower the pH in the sarcomere reaches. This potentially acts as a further checkpoint to increase binding of S100A1 under conditions where damage could occur, decreasing the amount of N2A that would be binding to actin.

There is a possible alternative model that also warrants consideration. The binding affinity between N2A and F-actin has not been established but based on the existing data, it would be predicted to be in the low μM to high nM range (Dutta et al., 2018). This is in line with the binding strength between S100A1 and UN2A. It is worth considering that the N2A–F-actin interaction is actually not desirable, and this interaction exists simply to prevent that binding from occurring. Based on the existing data, the binding of N2A to F-actin becomes stronger with higher $Ca^{2+}$ concentrations, so it is possible that this interaction evolved, so that it would become stronger as conditions favored binding of N2A to actin. We favor the first model but can not discount the second model with the existing data.

### Implications of S100A1 binding

The N2A region is a central signaling hub in the extensible region of titin. There are a series of signaling proteins that have been shown to bind in this region. These include the MARP proteins, which act as transcription modulators; CAPN3, which is a stretch-sensing protease; and Smyd2, subsequent to the methylation of Hsp90 (Miller et al., 2003; Sorimachi et al., 1995; Hayashi et al., 2008; Yamasaki et al., 2001; Ohashi et al., 2007; Voelkel et al., 2013; Donlin et al., 2012). In addition to signaling proteins, the N2A region has been shown to bind to actin, implicating the interaction in activated muscle (Donlin et al., 2012; Voelkel et al., 2013; Zhou et al., 2021). With all these interactions, this region of titin is a crowded location, and it is not clear when particular interactions occur. The addition of potential interactions from S100A1 makes understanding which interactions take precedence over other interactions an important question.

Identifying a binding site for S100A1 leads to potential comparisons with RYR, which is regulated by both CaM and S100A1 (Prosser et al., 2011; Wright et al., 2008; Treves et al., 1997; Xu et al., 2017). The binding sites for CaM and S100A1 on RYR appear to overlap, meaning that only one protein would be able to bind at a time (Prosser et al., 2008; Ivanenkov et al., 1995). Functionally, S100A1 increases the potential of the channel being open, while CaM inactivates the channel (Prosser et al., 2011; Woll et al., 2021; Rodney et al., 2000). Currently, the specific function of S100A1 binding to UN2A is not clear, but it is tempting to speculate that a similar competitive reaction might occur with N2A, as has been observed with the RYR. Further studies to determine if CaM binds to this region will be important in the future to understand how $Ca^{2+}$-binding proteins impact the function of the N2A region.

It is also possible that the $Ca^{2+}$-binding protein interaction is muscle type dependent. S100A1 is more highly expressed in the cardiac muscle, so it is possible that this binding interaction dominates in the cardiac muscle. S100A1 has been shown to disrupt the interaction of PEVK with actin in previous experiments. Since the N2A region has been shown to bind actin, it is possible that S100A1 binding might play a similar role with the N2A region. The PEVK segment has rich tandem repeats of glutamate residues, and UN2A's disordered regions are similar in the sense that it also contains glutamate-rich regions, which may be playing a role in S100A1 binding at lower pH.

### Summary

$Ca^{2+}$ plays an important role in muscle function and has been shown to regulate titin function, both directly and indirectly. The work presented here provides evidence that the $Ca^{2+}$-binding protein S100A1 interacts with the unique sequence in the N2A region of titin. The binding of S100A1 to UN2A becomes stronger as $Ca^{2+}$ concentrations increase, consistent with weak binding occurring when there is one $Ca^{2+}$ bound per subunit and binding becoming stronger when the second $Ca^{2+}$ ion is bound. The conformation of UN2A also appears to influence binding as decreased pH results in a more helical structure in UN2A and this correlates with stronger S100A1 binding. These results provide an intriguing new aspect to regulation of titin function in the active muscle.

### Data availability

Raw data are available from the corresponding author upon reasonable request.

## Acknowledgments

Henk L. Granzier served as an editor.

The authors would like to thank Dr. Kiisa Nishikawa for her initial discussions on the potential of calcium-binding protein interactions with this region of titin.

Author contributions: S.I. Apel: conceptualization, data curation, formal analysis, investigation, methodology, resources, software, supervision, validation, visualization, and writing—original draft, review, and editing. E. Schaffter: formal analysis, investigation, and writing—original draft, review, and editing. N. Melisi: data curation and formal analysis. M.J. Gage: conceptualization, data curation, methodology, project administration,

resources, swupervision, writing—original draft, review, and editing.

Disclosures: The authors declare no competing interests exist.

Submitted: 21 August 2023

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

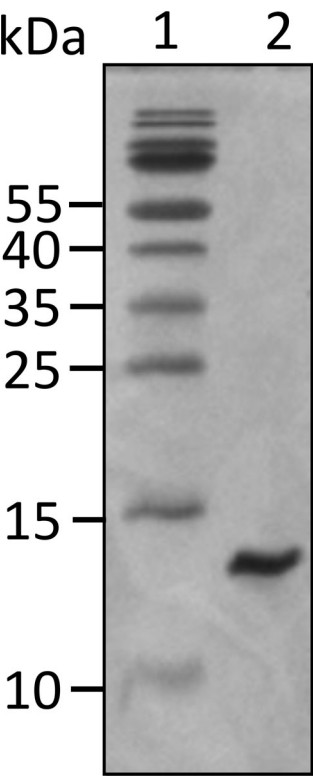

Figure S1. **16% SDS-PAGE gel for UN2A.** The titin gene sequence of the UN2A region was commercially synthesized by GeneArt (Invitrogen) and encompasses amino acids 8539 to 8655 in the mouse titin sequence (GenBank accession no. NM_011652.3), which shares 95% identity with the human UN2A sequence (111 out of the 117 amino acids are identical). Sequence for UN2A: DERKKQEKIEGDLRAMLKKTPALKKGSGEEEEIDIMELLKNVDPKEYEKYARMYGITDFRGLLQAFELLK QSQEEETHRLEIEELEKSERDEKEFEELVAFIQQRLTQTEPVTLIKD. Source data are available for this figure: SourceData FS1.

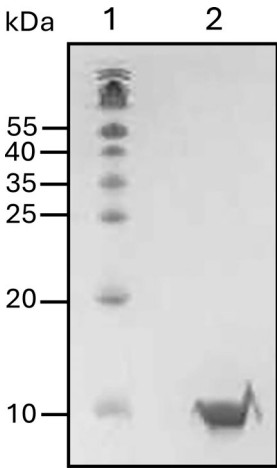

Figure S2. **16% SDS-PAGE gel for S100A1.** The mouse S100A1 gene sequence was graciously provided by David J. Weber's lab from the University of Maryland. (GenBank accession no. NM_011309.3) Sequence for S100A1: MGSELESAMETLINVFHAHSGQEGDKYKLSKKELKDLLQTELSGFLDVQKDADAVDKVMKELDENGDGEVDFKEYVVLVAALTVACNNFFWETS. Source data are available for this figure: SourceData FS2.

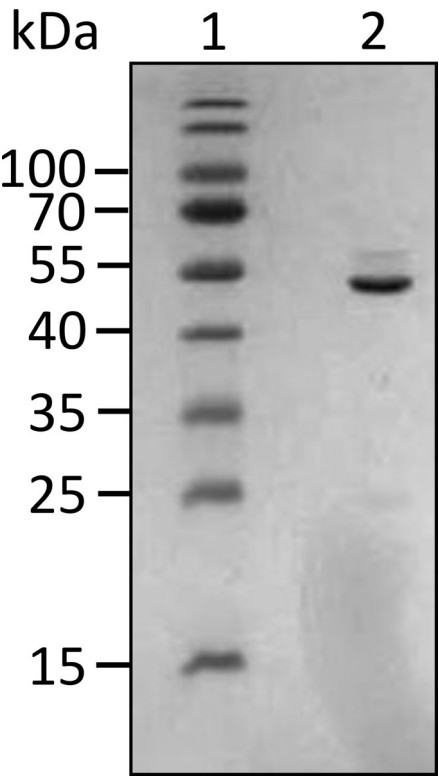

Figure S3.   **12% SDS-PAGE gel for UN2A-FRET.** The UN2A region was PCR amplified as described in Tiffany et al. (2017). The resulting PCR product was digested using NdeI and XhoI and ligated into the CFP-YFP containing vector described by Ohashi et al. (2007). Sequence for UN2A-FRET:HHHHHHMVSKGEELFTGVVPIL-VELDGDVNGHKFSVSGEGEGDATYGKLTLKLLCTTGKLPVPWPTLVTTLGYGVQCFARYPDHMKQHDFFKSAMPEGYVQERTIFFKDDGNYKTRAEVKFEGDTLVNRIELK-GIDFKEDGNILGHKLEYNYNSHNVYITADKQKNGIKANFKIRHNIEDGGVQLADHYQQNTPIGDGPVLLPDNHYLSYQSALFK DPNEKRDHMVLLEFLTAAGITEGMNELYK DERKKQEKIEGDLRAMLKKTPALKKGSGEEEEIDIMELLKNVDPKEYEKYARMYGITDFRGLLQAFELLKQSQEEETHRLEIEELEKSERDEKEFEELVAFIQQRLTQTEPVTLIKDMVSKGE-ELFGGIVPILVELEGDVNGHKFSVSGEGEGDATYGKLTLKFICTTGKLPVPWPTLVTTLTWGVQCFSRYPDHMKQHDFFKSVMPEGYVQERTIFFKDDGNYKTRAEVKFEGDTLVNR-IELKGIDFKEDGNILGHKLEYNYISHNVYITADKQKNGIKANFKARHNITDGSVQLADHYQQNTPIGDGPVILPDNHYLSTQSALSKDPNEKRDHMVLLEFVTAAGITHGMDELYK. Source data are available for this figure: SourceData FS3.

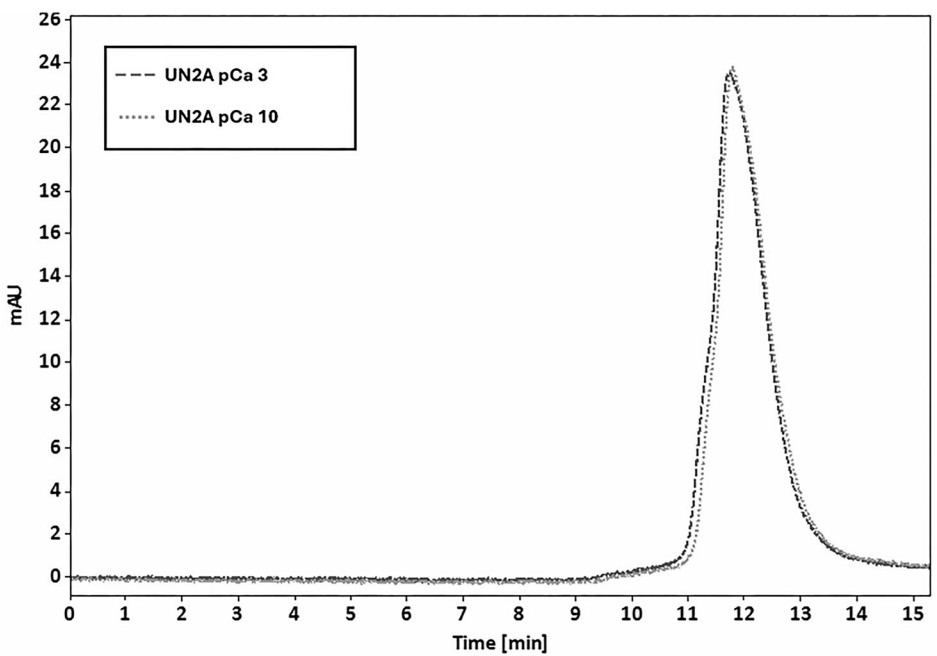

Figure S4.   **SEC chromatogram of UN2A with and without Ca²⁺.**

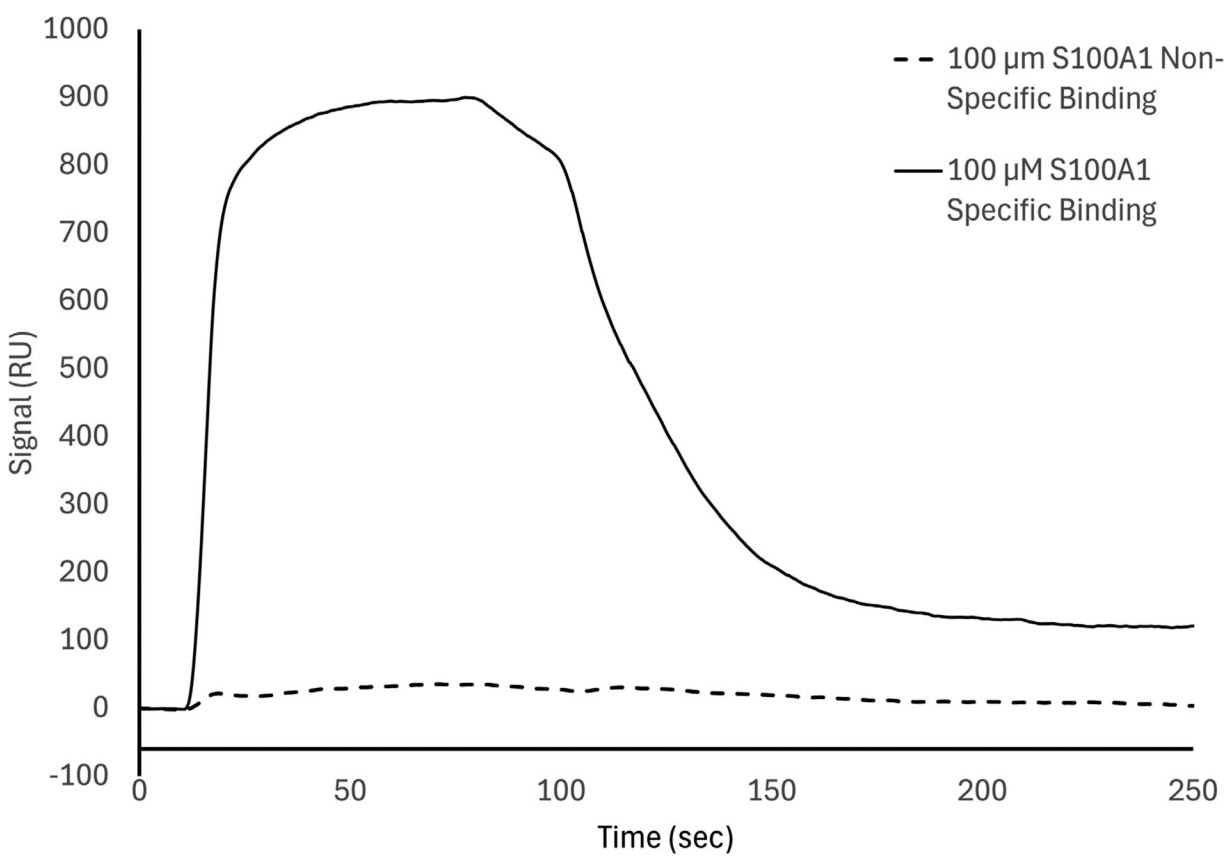

Figure S5.  **SPR nonspecific binding data.** SPR affinity one-to-one fits.

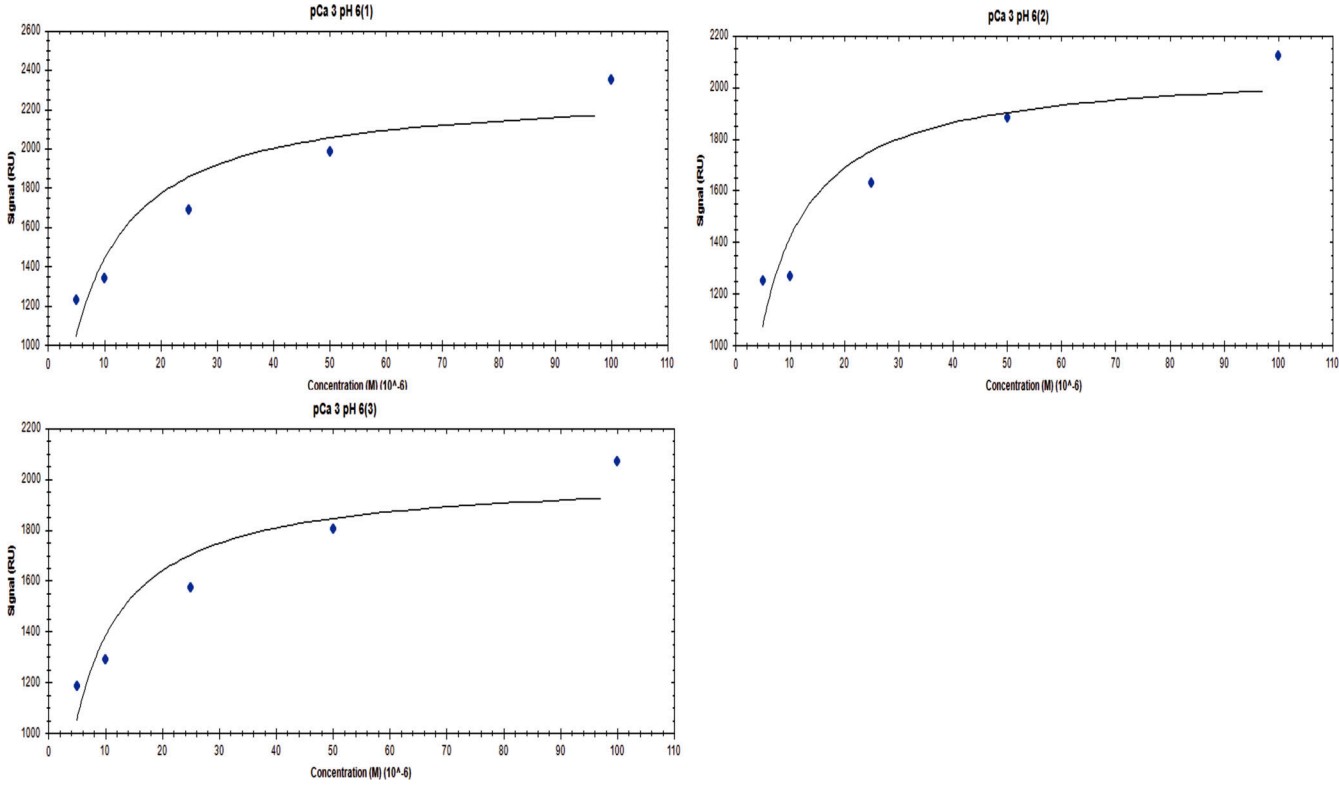

Figure S6.  **Affinity one-to-one fits for S100A1 binding to UN2A at pCa 3 and pH 6.**

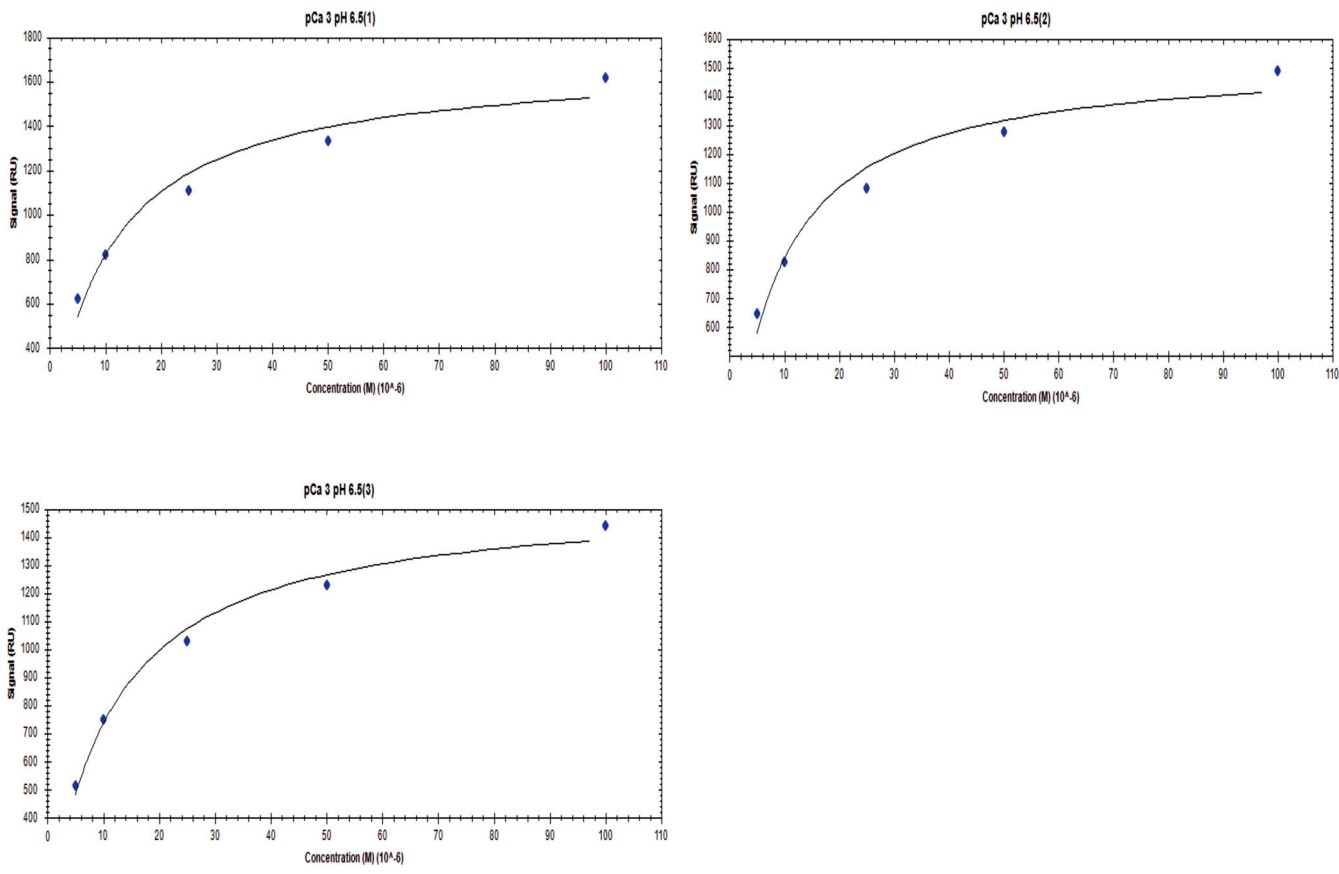

Figure S7.  **Affinity one-to-one fits for S100A1 binding to UN2A at pCa 3 and pH 6.5.**

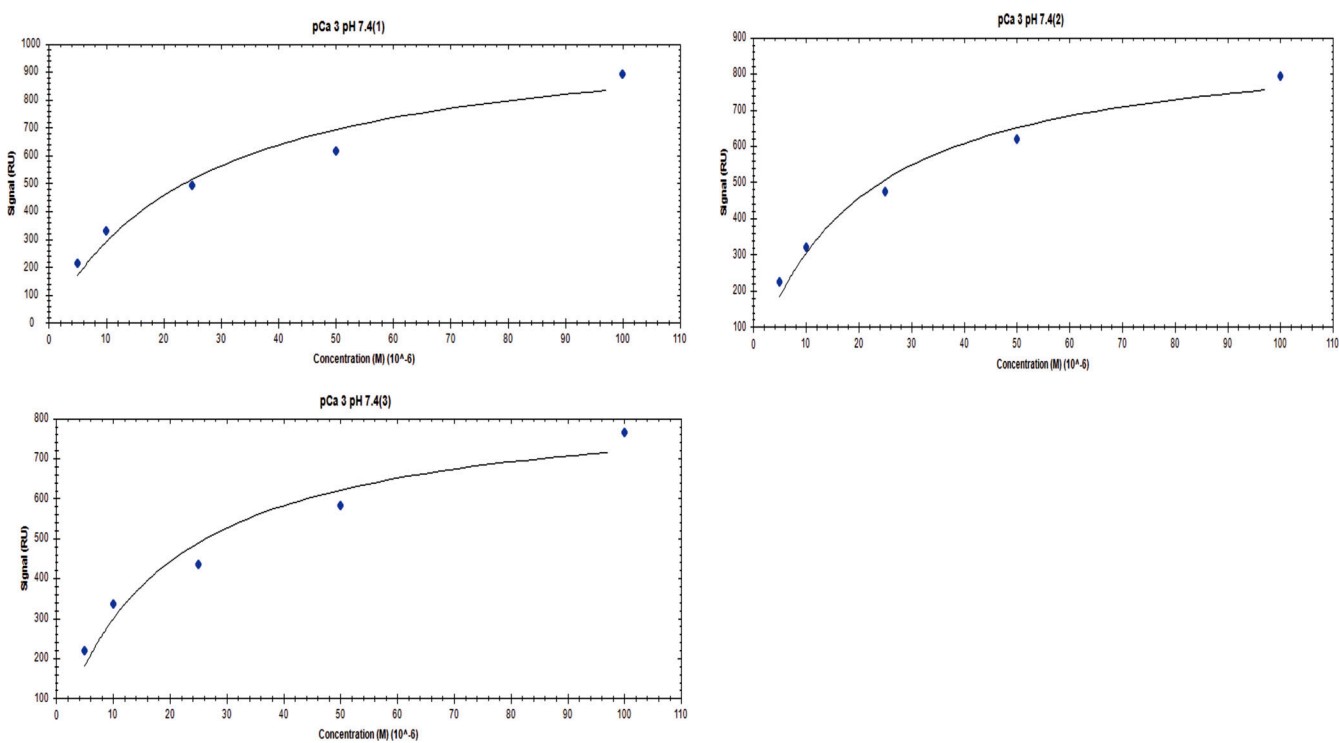

Figure S8.  **Affinity one-to-one fits for S100A1 binding to UN2A at pCa 3 and pH 7.4.**

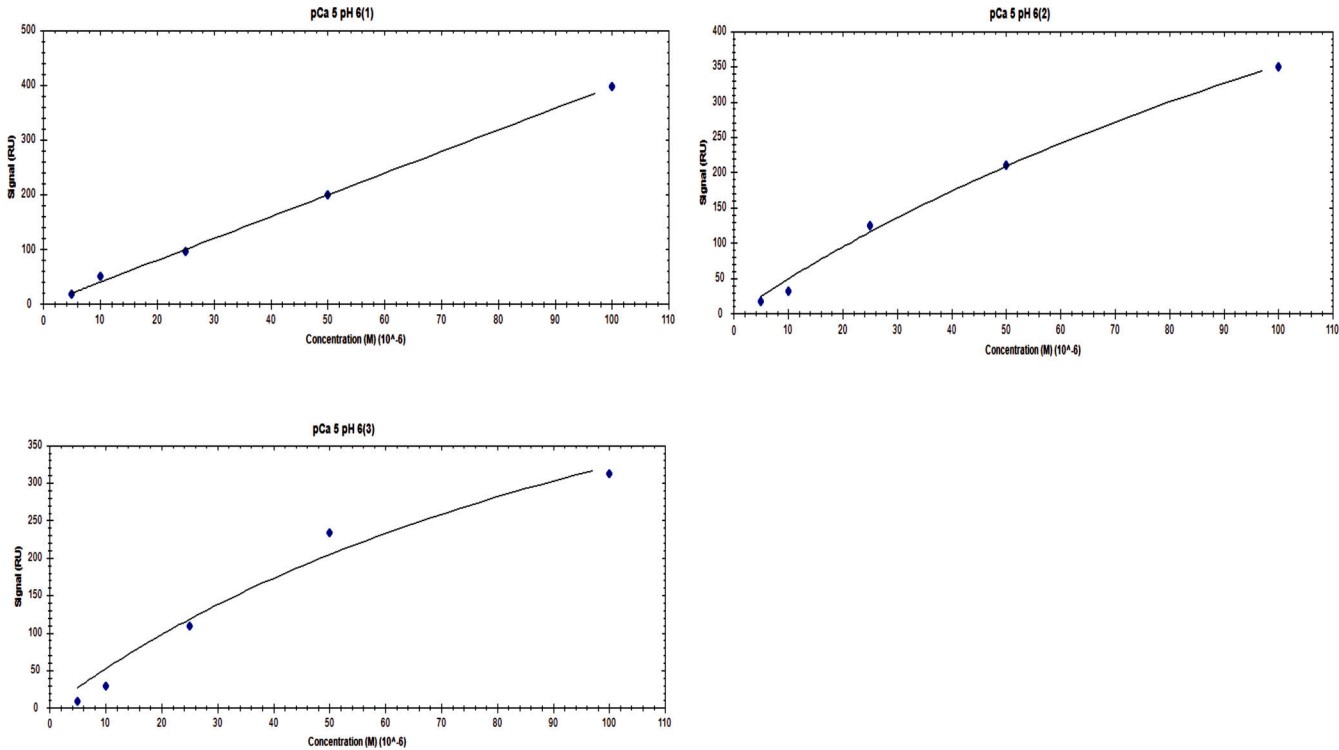

Figure S9.  **Affinity one-to-One fits for S100A1 binding to UN2A at pCa 5 and pH 6.**

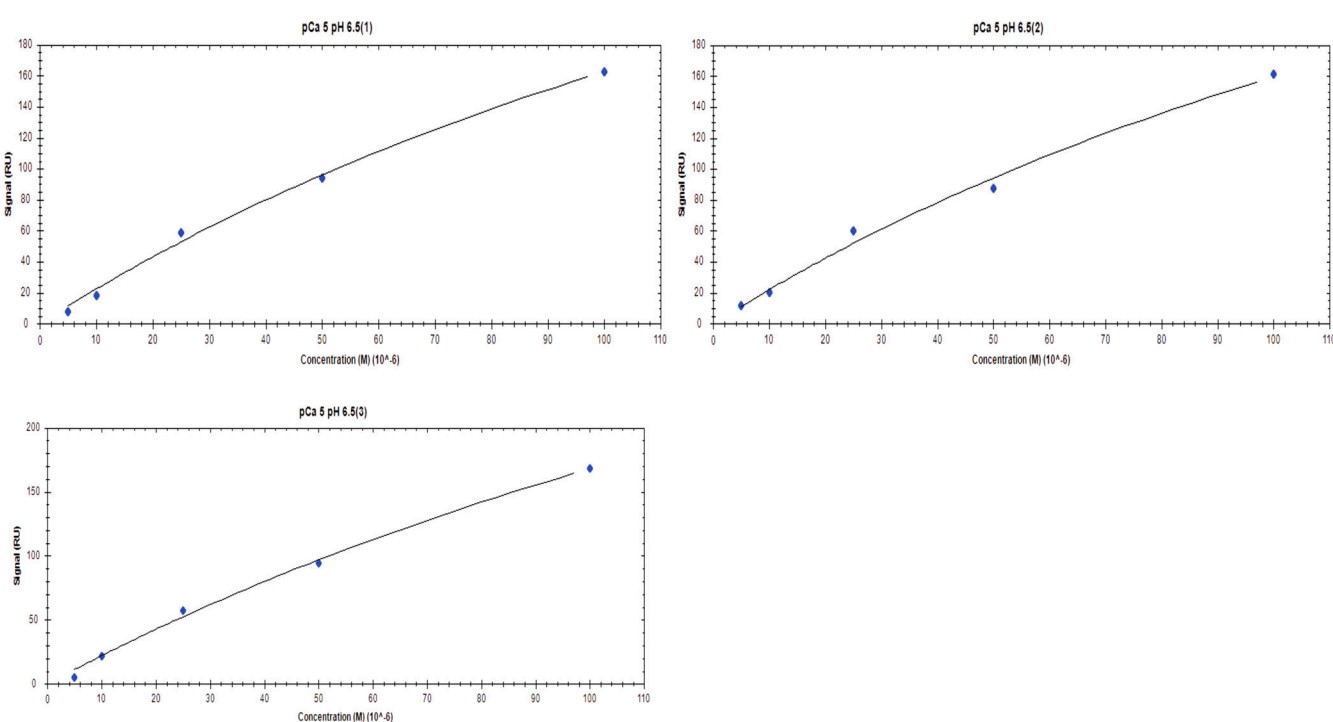

Figure S10.  **Affinity one-to-one fits for S100A1 binding to UN2A at pCa 5 and pH 6.5.**

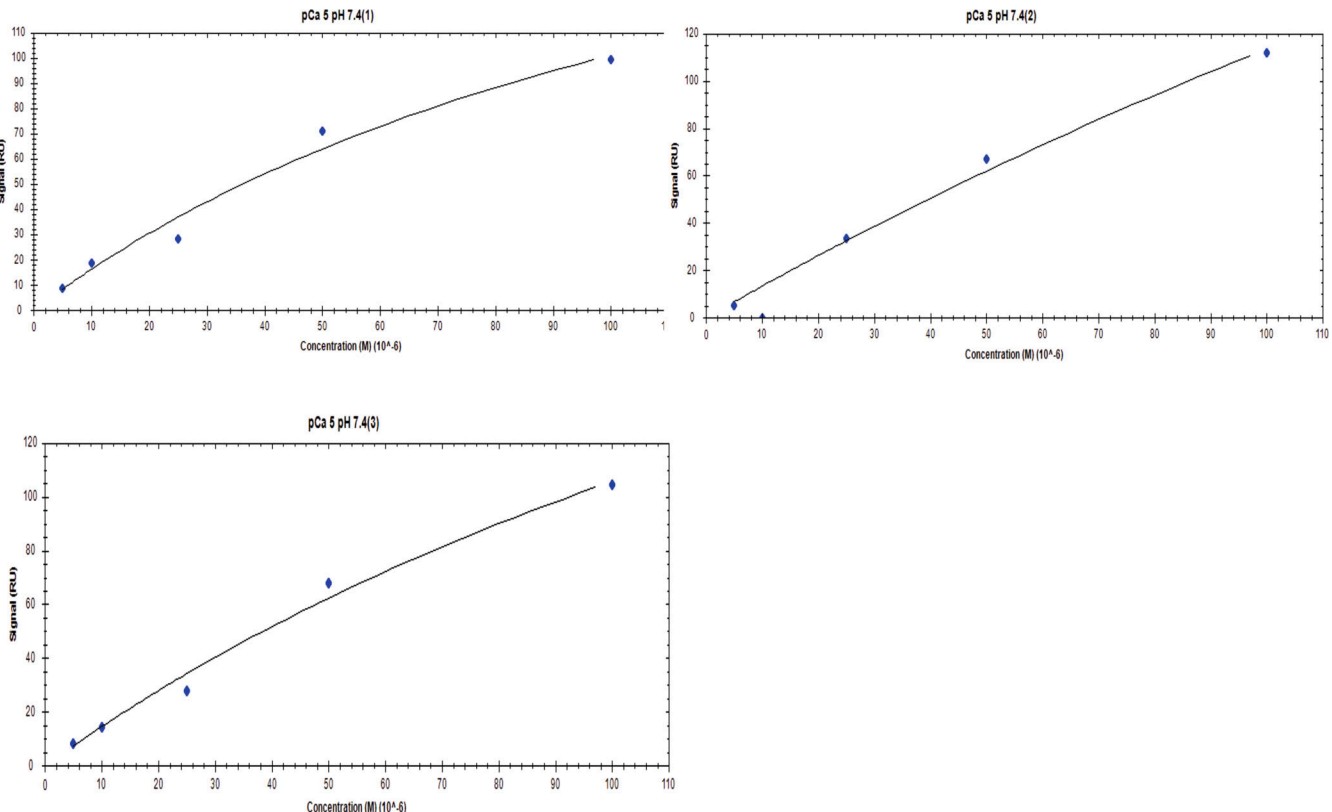

**Figure S11.** **Affinity one-to-One fits for S100A1 binding to UN2A at pCa 5 and pH 7.4.**

Provided online is Data S1. Data S1 shows RYR and UN2A alignment.

*Plasmid for UN2A*

The titin gene sequence of the UN2A region was commercially synthesized by GeneArt (Invitrogen, Grand Island, NY, USA) and encompasses amino acids 8539–8655 in the mouse titin sequence (NCBI accession number: NM_011652.3), which shares 95% identity with the human UN2A sequence (111 out of the 117 amino acids are identical).

*Sequence for UN2A*

DERKKQEKIEGDLRAMLKKTPALKKGSGEEEEIDIMELLKNVDPKEYEKYARMYGITDFRGLLQAFELLK
QSQEEETHRLEIEEELEKSERDEKEFEELVAFIQQRLTQTEPVTLIKD.

