## [Peer Review File · The Journal of General Physiology]

The Calcium-Binding Protein S100A1 Binds to Titin's N2A Insertion Sequence in a pH Dependent Manner

Sabrina Apel, Emily Schaffter, Nicholas Melisi, and Matthew Gage

Corresponding Author(s): Matthew Gage, University of Massachusetts Lowell and Sabrina Apel, University of Massachusetts Lowell

Review Timeline:

Submission Date:	August 21, 2023
Editorial Decision:	September 19, 2023
Revision Received:	July 18, 2024
Editorial Decision:	August 6, 2024
Revision Received:	October 21, 2024
Editorial Decision:	November 6, 2024
Revision Received:	November 13, 2024
Editorial Decision:	November 18, 2024
Revision Received:	November 26, 2024

Editor: Henk Granzier

Transaction Report:

DOI: <https://doi.org/10.1085/jgp.202313472>

September 19, 2023

Dr. Matthew J Gage
University of Massachusetts Lowell
One University Avenue
Lowell, MA 01854

Re: 202313472

Dear Dr. Gage,

Your manuscript entitled "S100A1 Binds to Titin's N2A Insertion Sequence in a pH Dependent Manner" has now been seen by 3 reviewers, whose comments are appended below. You will see that the reviewers have raised concerns about the manuscript, including applying appropriate controls, showing repeatability of results, and addressing the physiological significance of your findings.

As a result of these concerns, I am afraid that we cannot offer to publish your manuscript. However, if you believe that further experiments would allow you to address these issues to the satisfaction of the editors and reviewers, we would be willing to consider a substantially revised manuscript if resubmitted within one year. In the case of eventual publication, the article would include a 'revised date' alongside its submitted and accepted dates. In the event that the revision process takes longer, any resubmitted manuscript would be treated as a new submission and would be given a new manuscript number. Either way, any revised manuscript would be sent out for review to the original reviewers, subject to their availability, and we must emphasize that we cannot predict the final outcome.

Should you decide to submit a revised version, please submit your revised manuscript via the link below and include a letter that details your responses to the editors' and reviewers' concerns. Also, to facilitate evaluation of revised manuscript by Reviewers and Editors, please provide a copy of the text with alterations highlighted (e.g., boldfaced, underlined, or in color). If you feel changes are too extensive to provide a version of your paper with changes highlighted, please state so in your response.

Please pay particular attention to recent changes to our instructions to authors in the following sections: Data presentation, Blinding and randomization and Statistical analysis, under Materials and Methods, as shown here: <https://rupress.org/jgp/pages/submission-guidelines#prepare>. Re-review will be contingent on inclusion of the required information (including for data added during revision) and demonstration of the experimental reproducibility of the results. Also, to improve the reproducibility of published content, we have partnered with SciScore. Authors are prompted in eJP to copy and paste the Materials and Methods section of their manuscript for a SciScore assessment when submitting their revised manuscript. Authors are encouraged (not required) to further revise their Materials and Methods if the SciScore is below 4. More information can be found here: <https://rupress.org/jgp/pages/submission-guidelines#sciscore>

Please do not hesitate to contact me (via the editorial office) if you feel that a discussion of the reviewers' and editors' comments would be helpful.

Thank you for the opportunity to consider your manuscript, a copy of which we will retain for our files.

Sincerely,

Henk L. Granzier, Ph.D.
On behalf of Journal of General Physiology

Journal of General Physiology's mission is to publish mechanistic and quantitative molecular and cellular physiology of the highest quality; to provide a best-in-class author experience; and to nurture future generations of independent researchers.

Please submit your revised manuscript via this link within one year:
Link Not Available

Reviewer #1 (Comments to the Authors):

This paper investigates the binding of the calcium-responsive S100A1 protein to N2A using analytical gel filtration. They then use SPR to characterise the strength of the binding, which they show to be weak but responsive to both pCa²⁺ and pH.

Furthermore, they use FRET to determine conformational changes of the N2A due to S100A1 binding and pH. They propose that S1001A binding can act as a sensor.

This is a very interesting proposed interaction at the N2A which adds to a growing body of literature on its importance as a signalosome in muscle. I am keen to see this shown and this manuscript published, however, I have the following concerns about the strength of the manuscript and the claims being made:

It is not detailed what construct/definition of the UN2A is being used. The residue range as relates to a published version of titin is required (the full canonical transcript is often used).

Perhaps an orientation intro figure with a schematic and the tri-helical bundle shown and the residue ranges annotated for people not familiar with the N2A, something similar for the S1001A would also be nice. The structure of the THB and S1001A are known so could be shown. At the moment I feel it would be difficult for someone not so familiar with titin/ N2A to know what is going on.

For completeness, the intro should mention that MARP1 is known to facilitate titin binding to actin.

SPR is good for calculating affinities and kinetics of known binders but due to the complications of non-specific binding it should not be used as evidence of binding (just for the characterisation of known binders). See below for specific concerns about the SPR.

This means that the analytical SEC needs to be completed to a higher standard and ideally a second method to show the complex interaction unambiguously. It would be nice to have some sort of conformation that this complex can exist in cells, even colocalization in a non-muscle cell line would be interesting. If not possible possibly the authors could utilise the split fluorescence method they use in this paper but with the split across the two partners rather than in the same protein. A simple pull-down would also suffice.

The gel filtration as currently performed is wholly unconvincing.

The chromatogram of each individual protein needs to be shown- at the moment we are not even shown that the 'complex' elutes at a lower retention volume than the individual proteins, we are only told that this is the case.

The fractions of the gel filtration also need to be shown in SDS PAGE for each chromatograph of the two proteins alone and then together. A clear co-migration of the two proteins should be visible.

I know this presents a technical difficulty as both proteins have the same MW - however, this can be overcome by putting a large tag (GST is common) on one of the proteins (of course the tag would then also have to be added to the partner protein alone as a non-binding control). Another option that would not require further controls would be just to include one of the flanking Ig domains to the UN2A, thus making the protein larger. Once the interaction has been unambiguously shown, then the smaller constructs can be investigated to discover the minimal binding region/ calcium's effects on the binding. At the moment this result cannot be disambiguated from increased hetero-oligomerisation/aggregation.

Also, by showing no gels we currently have no means to assess the purity of the proteins to determine if the association could be from or promoted by a contaminating partner (this is also an issue with the SPR).

Other ways of confirming the specific interaction would be site-directed mutagenesis to destroy the complex formation, this would have an added benefit of localising the binding site. In line 221 the authors say they identified a weak homology with a consensus sequence - this would be nice to show in a figure. It should also be tested via mutagenesis.

Minor- What are the exclusion volumes of each peak?

SPR

The SPR as it is currently performed is missing several controls and therefore the results are over-interpreted.

Are the SPR graphs the original traces or fitted curves? The curves look very smooth which makes me suspect the latter. Either way, both should be shown to allow an assessment of the fit. The raw data should be given in supplementary.

The graphs would benefit from vertical lines which part of the experiment is being shown i.e. the association and dissociation steps. Although not always needed I think it is here as I am having to guess when the dissociation step occurred due to the unusual shape of the curve (especially in Fig 8).

A negative control to show the amount of non-specific binding should be shown - this should be the maximal protein concentration over the unloaded chip (with and without calcium). If non-specific interactions are causing the curves seen then adding more calcium would likely decrease the signal.

As the SPR experiments are currently performed they are not rigorous enough to offer a K_d . This is due to the following issues:

The low affinity seen could be due to the coupling method to the chip, as biological meaning is derived from this feature this possibility should be mentioned.

As direct coupling was used it is even more important to provide an SDS PAGE with the purified protein shown as all proteins in the mixture would be bound, it is critical to know that it was a pure sample.

At high concentrations the dissociations are incomplete. Several of the titrations do not look like they have reached equilibrium.

As the binding is not monovalent quantitative analysis should ideally not be attempted. However, if the authors wish to do this the titration should be done in both directions (increasing the concentration of the binding partner and then decreasing concentration and the K_d s compared - they should be the same).

The authors also need to determine if the 2-mode binding could be real or is in fact an artefact of the way the protein is bound to the chip. To rule this out these controls are required:

S1001A to the chip and UN2A titrated (if this behaviour disappears, it is an artefact)

An alternative would also be using an indirect immobilisation method using a tag on the protein.

Ideally, to be robust these controls should also be performed:

2 independently produced batches of protein

2 different recombinant forms of protein (i.e. slightly different constructs).

The Scatchard plots should be shown.

The effects of the calcium could be due to conformational changes in the S1001A or electrostatic screening -could this be disambiguated by mutating the Ca^{2+} coordination residues of S100A1

Minor- the order of the legend in fig 2 is in the wrong order

Line 249 - nothing is shown or numbers given about the goodness of the model fit

Smaller issues:

CD - no cuvette size is given

Methods state data from 190-250 but this has been cut to 200 nm in the figure (understandable because of the Tris buffer causing noise under 200nm but the discarded data should be mentioned).

Fig 4, 9A and 9B legend- what is IS-F ? is it N2A-IS what does the F stand for?

Line 171 - what is the N2A-IS is this the same as the UN2A?

Also, this method describes just one protein but the figure legend of Fig 1 and the text indicate Un2A and S1001A. How was the binding study performed? What was the stoichiometry of the mixture? What was the incubation time and buffer?

I am not familiar enough with the split fluorescence procedure to judge the appropriateness of this method and results but this appears to be ok. The CD, although quite jittery, is convincing to me, so I think these results and conclusions make sense.

Note: Nomenclature- the Igs around the N2A are actually called I80, I81 etc not Ig80 as the 'I' stands for I-band not Ig.

<https://www.ahajournals.org/doi/10.1161/hh2301.100981>. Although this error has been made several times before in the literature and names at the end of the day don't matter as long as the gene details are there (see below).

No details of the expression plasmid for any constructs are given, this is important information to know what the cloning artefacts will be and to aid reproducibility. All genes need to be referenced as to what they are and the residue ranges.

The concentration of the antibiotics used is not given- this is also important for reproducibility. The full names of the antibiotics should also be used so that those not familiar with the methods can reproduce them. AMP and Chlor is 'lab speak'.

Line 127- 10,000 RPM - either give the rotor as well or use g - currently this information is incomplete and not reproducible.

Line 156

Are the acronyms CFP and YFP widely known?

Line 291 - please add the reference to your previous work

My final thoughts are it would be nice to discover if the S100A1 binding site is as extensive/ competes with the MARP1 site which extends to the I81 domain. Although perhaps this is beyond the scope of the manuscript.

Reviewer #2 (Comments to the Authors):

The manuscript JMCC-202313472 "S100A1 Binds to Titin's N2A Insertion Sequence in a pH Dependent Manner" by Apel et al characterizes a new binding site for S100A1. The interaction between S100A1 and titin's PEVK region is well known. The current study utilizes recombinant proteins, Size Exclusion Chromatography, Circular Dichroism, FRET, and Surface Plasmon Resonance to show that S100A1 does bind to the N2A insertion sequence, that it changes titin's conformation, and that this conformational change interaction can be regulated by pH.

Reviewer Comments:

1. The author's discussion analyzes the implications of S100A1 binding (Line 422+), where cooperative or potentially competitive binding are discussed. However, function is not discussed. Do the conformational changes potentially relate to changes in muscle passive tension or other titin-related physiologic functions?
2. Binding Lifetime/Kinetics: The authors suggest that the weak binding may be helpful in a physiological context, and potentially differential by tissues. Since a major difference in muscle activation is the twitch activation of cardiac cells versus the twitch and possible tetanic activation of the skeletal muscles, would the binding lifetime be a significant physiologic factor?
3. Methods- Plasmids:
 - 3a. The plasmid sequences should be included in a supplement or reference for reproducibility.
 - 3b. It is unclear from the description of the UN2A whether flanking Ig domains may be included.
4. Results (Lines 220-222): The results describe a sequence analysis of the UN2A region and binding homology. This result may need to be described in the methods and the homology shown for rigor and reproducibility purposes. A reference to a previously published comparison between S100A1 and the UN2A region would be sufficient.

Reviewer #3 (Comments to the Authors):

I congratulate the authors on their S100A1-N2A manuscript. This work provides new insights into the titin signalosome surrounding the N2A element, which has been shown to regulate various processes in striated muscle. Although I like the work presented by Apel et al, after critically reviewing the manuscript I have listed several suggestions below that would benefit the manuscript.

Major/experimental revision

There is quite a bit of discussion on sequence analysis for S100A1 binding in the result section. I would like to see this analysis that compares the RYR1/2 S100A1 binding sequence to titin's N2A and PEVK sequence. Perhaps comparing among species can yield some further insights. Or if interesting, comparisons of CaM or Casq binding-sites.

I miss a potential role for S100A1 interacting with N2A. As interaction with fActin is frequently discussed, I'd like to see fActin co-sedimentation assays with UN2A and S100A1 at physiological pH and pH 6. Ideally I'd also like to see these experiments with mono- and di-calcium bound S100A1.

The UN2A is also known to be phosphorylated by PKA (Lanzicher et al 2020), does this phospho-site affect S100A1 binding? Your FRET assay seems ideal for probing the role.

Do larger N2A constructs also interact the same with S100A1? UN2A-I81 or I80-UN2A might have more structural stability and bind S100A1 under more physiological conditions.

Major/text revision

Your discussion would greatly benefit from moving away from physiological conditions for the S100A1-N2A interaction. The relatively extreme conditions needed for S100A1 to bind to N2A make this mechanism unlikely under normal/healthy conditions. Focusing on extreme exercise and muscle damage makes more sense in the context of your data. In addition, how would the broader S100 family of proteins contribute and are they likely to bind to N2A?

Minor/text revision

Introduction

Line 44; I suggest removing the approximate size at 4.0 mDa. Titin can range anywhere from 0.6-4.0 mDa depending on splicing.

Line 46; Titin is an important contributor of passive tension to muscle. Describing this as passive force enhancement is confusing and I advise rewriting this sentence.

Line 65-66, there is a new structure for N2A released recently by Stehle et al

Line 68-69; MARP is a family of 3 proteins, see Miller et al 2003.

Line 75-76; N2A interacting with Ca²⁺ has been proposed by several groups. Interaction with fActin-N2A has been published by several groups, with the role of Ca²⁺ being ambiguous, some showing enhancement others showing no effect.

Methods

Line 121; the UN2A construct requires more description it is unclear if this is just the coiled-coil or the whole unique sequence.

Line 137; What was the source for the S100A1 plasmid, and which sequence was used.

Results

Although I appreciate single graph pages for reviewing the manuscript, I do think the graphs can be condensed and combined into fewer figures to aid the narrative.

Discussion

Line 333; same comment as in the introduction, several groups have looked at this interaction between N2A and fActin. MARP1 also seems to modify N2A-fActin binding in a calcium based mechanism (Zhou et al 2021 and Van Der Pijl et al 2021).

Considering the MARP proteins possess Casq-binding sites, have you considered interaction of the MARPs and S100A1?

Line 336; titin kinase is not dependent on CaM, as shown by Lange 2005 Nature. The predicted CaM site in the kinase domain currently has no known relevance and even the kinase itself has disputed activity. There are several papers on the role of Ca²⁺ interacting with the PEVK segment of titin that should be cited and discussed here.

Line 353/354; Please provide citations for the physiological pH and pCa in striated muscle.

Line 374; What did your SEC data show as far as mono-, di- or oligomeric binding of S100A1 to UN2A? Did calcium concentration change the conformation of UN2A by itself (neutral pH)?

Line 406; Extended muscle activation only minimally increases the pool of intracellular calcium and tetanic contraction rarely happens in skeletal muscle.

Line 424-426; This segment is poorly written and misdescribes the functions of the N2A-binding proteins: MARPs are not known as transcription factors. MARPs are adapter/chaperone proteins that can interact with transcription factors. Smyd2 is a methyltransferase that can methylate HSP90ab, a PTM which makes Hsp90 bind to N2A. I suggest looking up several recent reviews on N2A and interacting proteins.

We would like to thank the reviewers for their careful and critical feedback on our manuscript. This feedback has resulted in a significantly improved manuscript, and we look forward to hearing feedback on our revisions. A point by point description of how we addressed the reviewer comments is listed below.

Reviewer #1:

It is not detailed what construct/definition of the UN2A is being used. The residue range as relates to a published version of titin is required (the full canonical transcript is often used).

We have included the requested details of both the UN2A and S100A1 sequences used in both the methods section and the supplemental data section which we are including with the revised manuscript.

Perhaps an orientation intro figure with a schematic and the tri-helical bundle shown and the residue ranges annotated for people not familiar with the N2A, something similar for the S1001A would also be nice. The structure of the THB and S1001A are known so could be shown. At the moment I feel it would be difficult for someone not so familiar with titin/ N2A to know what is going on.

We have included an introductory figure to help orient those who might not be as familiar with titin's structure. We have also included a structure of S100A1 in this figure as well.

For completeness, the intro should mention that MARP1 is known to facilitate titin binding to actin.

We have included the requested information about MARP1 in the third paragraph of the introduction.

SPR is good for calculating affinities and kinetics of known binders but due to the complications of non-specific binding it should not be used as evidence of binding (just for the characterisation of known binders). See below for specific concerns about the SPR.

This means that the analytical SEC needs to be completed to a higher standard and ideally a second method to show the complex interaction unambiguously. It would be nice to have some sort of conformation that this complex can exist in cells, even colocalization in a non-muscle cell line would be interesting. If not possible possibly the authors could utilise the split fluorescence method they use in this paper but with the split across the two partners rather than in the same protein. A simple pull-down would also suffice.

-also-

The gel filtration as currently performed is wholly unconvincing. The chromatogram of each individual protein needs to be shown- at the moment we are not even shown that the 'complex' elutes at a lower retention volume than the individual proteins, we are only told that this is the case.

-also-

I know this presents a technical difficulty as both proteins have the same MW - however, this can be overcome by putting a large tag (GST is common) on one of the proteins (of course the tag would then also have to be added to the partner protein alone as a non-binding control). Another option that would not require further controls would be just to include one of the flanking Ig domains to the UN2A, thus making the protein larger. Once the interaction has been unambiguously shown, then the smaller constructs can be investigated to discover the minimal binding region/ calcium's effects on the binding. At the moment this result cannot be disambiguated from increased hetero-oligomerisation/aggregation.

We have increased the robustness of the gel filtration data that we have included. Our initial studies were conducted using a Superdex 75 column on an Akta system, and the point that this was unconvincing is valid. We have repeated these experiments using our HPLC to provide a more robust characterization of the binding and have revised Figure 2 and included an additional figure. The suggestion of using the UN2A-FRET clone to separate the retention times of the isolated UN2A and S100A1 provided a more convincing demonstration of the shift due to binding from the S100A1. Figure 2 shows the chromatograms of the two individual proteins, which now differ in retention time by ~1 minute. The complex has a maximum retention time that is 0.1 minutes faster than the UN2A-FRET clone. This is a small shift, but a large shift is not expected since the UN2A-FRET clone is now significantly larger than the S100A1 and a large shift in Stokes Radius is not expected. While small, this shift is reproducible and does not appear in the absence of calcium.

We have also included a new figure that shows the impact of shifts in S100A1 concentration. As can be seen, there is not a change in retention time of the UN2A-FRET at pCa 10, where there is no calcium bound to the S100A1. However, there is an increase in the retention time shift as S100A1 concentration is increased, as would be expected if there is a concentration dependent equilibrium between the S100A1 and UN2A. This provided additional evidence for binding between S100A1 and UN2A.

Minor- What are the exclusion volumes of each peak?

We have included the retention times for the peaks in the SEC data.

The fractions of the gel filtration also need to be shown in SDS PAGE for each chromatograph of the two proteins alone and then together. A clear co-migration of the two proteins should be visible.

While this would be ideal, our HPLC is not equipped with a fraction collector and there is not a reliable way to correlate manually collected fractions with specific peaks. In addition, the samples we are using are 20 μ L injections so the sample is too dilute when it comes off the column for gel analysis.

Also, by showing no gels we currently have no means to assess the purity of the proteins to determine if the association could be from or promoted by a contaminating partner (this is also an issue with the SPR).

We have included gels of the purified protein in the Supplemental data to demonstrate the level of purity of each protein.

Other ways of confirming the specific interaction would be site-directed mutagenesis to destroy the complex formation, this would have an added benefit of localising the binding site. In line 221 the authors say they identified a weak homology with a consensus sequence - this would be nice to show in a figure. It should also be tested via mutagenesis.

We have included more details about the homology to the consensus binding site, along with the alignment of the UN2A region to the Ryanodine receptor to demonstrate that this binding site has a non-consensus sequence. While mutagenesis would be ideal to confirm the specific binding site, that is beyond the scope of this paper, which is focused on the nature of the binding and not the specific binding sequence. We do intend to probe the specific binding sequence in future studies.

SPR

The SPR as it is currently performed is missing several controls and therefore the results are over-interpreted.

We have included a more detailed description of the controls used to validate our results into the methods section of the paper. Some of the highlights include that:

- The ligand was immobilized on channel 2 of the sensor with channel 1 being used to control for non-specific binding.
- The data presented in the paper is the corrected raw data signal with the non-specific binding removed.
- We performed both a baseline reference correction and non-specific binding subtraction on the data that has been included in the manuscript.

Are the SPR graphs the original traces or fitted curves? The curves look very smooth which makes me suspect he later. Either way, both should be shown to allow an assessment of the fit. The raw data should be given in supplementary.

The presented graphs are the subtracted original traces. Due to the fast dissociation, an affinity fit protocol was used to fit the data, using the signal from the binding plateau and fitting that at a range of concentrations (TRACEDRAWER 1.8 Handbook. <https://nicoyalife.com/wp-content/uploads/2018/04/TraceDrawer-1.8-Handbook.pdf>) All the fits from the individual trials are provided in the supplemental data and we have included a representative fitting curve in the manuscript to help show the methodology that was used.

The graphs would benefit from vertical lines which part of the experiment is being shown i.e. the association and dissociation steps. Although not always needed I think it is here as I am having

to guess when the dissociation step occurred due to the unusual shape of the curve (especially in Fig 8).

We have modified the traces to show where the association and dissociation steps are occurring on the traces we have included in the main body of the manuscript.

A negative control to show the amount of non-specific binding should be shown - this should be the maximal protein concentration over the unloaded chip (with and without calcium). If non-specific interactions are causing the curves seen then adding more calcium would likely decrease the signal.

Our instrument has two channels, so we collect non-specific binding simultaneously with the binding curves. We have included a figure in the supplementary data showing the binding curve and non-specific binding curve for 100 μ M S100A1 on the same plot.

As the SPR experiments are currently performed they are not rigorous enough to offer a K_d . This is due to the following issues:

The low affinity seen could be due to the coupling method to the chip, as biological meaning is derived from this feature this possibility should be mentioned.

Direct coupling may under some circumstances affect the structure/binding affinity of a protein. More likely is that the R_{max}/B_{max} is lower as the protein is randomly oriented on the surface (which would not affect affinity). However, direct immobilization is a standard SPR practice, and the purpose of the study is to compare affinity of the same immobilized protein under different conditions. We see a similar weak binding in our SEC experiments, supporting the conclusion reached by the SPR experiments. That said, we added a mention in the discussion that the coupling could be having some impact on the binding to acknowledge that we can't rule out this as a reason for the weak observed binding.

As direct coupling was used it is even more important to provide an SDS PAGE with the purified protein shown as all proteins in the mixture would be bound, it is critical to know that it was a pure sample.

As mentioned above, we have included SDS-PAGE gels for each of the proteins so show the level of purification we were able to obtain.

At high concentrations the dissociations are incomplete. Several of the titrations do not look like they have reached equilibrium.

Equilibrium is required for affinity measurement not kinetic measurement. While the association could have been allowed to progress longer, this should not affect how the kinetic data is evaluated, especially since this is a comparative study, and the conditions were kept constant.

As the binding is not monovalent quantitative analysis should ideally not be attempted. However, if the authors wish to do this the titration should be done in both directions (increasing the concentration of the binding partner and then decreasing concentration and the K_d s compared - they should be the same).

Since regeneration was performed, no carry-over effects should be present. All binding is being tested against the same non-bound ligand surface. We have tried to attach the S100A1 using several different types of chemistry and have not been able to couple the S100A1 to a chip to perform the binding experiments and similar challenges have been reported in other S100A1 studies (references?).

The authors also need to determine if the 2-mode binding could be real or is in fact an artefact of the way the protein is bound to the chip. To rule this out these controls are required: S100A1 to the chip and UN2A titrated (if this behaviour disappears, it is an artefact) An alternative would also be using an indirect immobilisation method using a tag on the protein.

We concede that the 2-mode binding is not supported by a strong justification beyond some of the data fitting better with this model. It is possible to explain as the S100A1 dimer dissociating from each other on the protein, but this is pure speculation at this point. We have removed this from the manuscript and have fit the data as a single binding mode in the absence of experimental evidence to the contrary. The proposed experiment would help address this but the challenges attaching S100A1 to the chip (see above) precludes this experiment from being performed.

Ideally, to be robust these controls should also be performed: 2 independently produced batches of protein

All experiments were replicated using multiple expression batches and using multiple chips to ensure robustness in the results.

The effects of the calcium could be due to conformational changes in the S100A1 or electrostatic screening -could this be disambiguated by mutating the Ca²⁺ coordination residues of S100A1.

This is an involved series of studies to explore this possibility and would be the focus of future studies.

Minor- the order of the legend in fig 2 is in the wrong order.

We have revised this in the legend.

CD - no cuvette size is given

The cuvette size has been added.

Methods state data from 190-250 but this has been cut to 200 nm in the figure (understandable because of the Tris buffer causing noise under 200nm but the discarded data should be mentioned).

The discarded data has been mentioned in the methodology, and the data shown is stated in the figure legend.

Fig 4, 9A and 9B legend- what is IS-F ? is it N2A-IS what does the F stand for?

The legend was changed for each figure. IS-F stood for the UN2A-FRET protein discussed in the manuscript.

Line 171 - what is the N2A-IS is this the same as the UN2A?

We have changed all references to N2A-IS to UN2A in the manuscript.

Also, this method describes just one protein but the figure legend of Fig 1 and the text indicate Un2A and S1001A. How was the binding study performed? What was the stoichiometry of the mixture? What was the incubation time and buffer?

We have added this information into the size exclusion chromatography methodology.

Note: Nomenclature- the Igs around the N2A are actually called I80, I81 etc not Ig80 as the 'I' stands for I-band not Ig. <https://www.ahajournals.org/doi/10.1161/hh2301.100981>. Although this error has been made several times before in the literature and names at the end of the day don't matter as long as the gene details are there (see below).

Thank you for pointing this out and we have revised the manuscript to correct this throughout the manuscript.

No details of the expression plasmid for any constructs are given, this is important information to know what the cloning artefacts will be and to aid reproducibility. All genes need to be referenced as to what they are and the residue ranges.

We have added the details for each expression plasmid in the methods and supplementary page.

The concentration of the antibiotics used is not given- this is also important for reproducibility. The full names of the antibiotics should also be used so that those not familiar with the methods can reproduce them. AMP and Chlor is 'lab speak'.

We've added the concentrations of the antibiotics, and have added the full names of the antibiotics to the manuscript.

Line 127- 10,000 RPM - either give the rotor as well or use g - currently this information is incomplete and not reproducible.

We have revised the RPM to xg

Line 156 - Are the acronyms CFP and YFP widely known?

We've added these to the abbreviations.

Line 291 - please add the reference to your previous work

Our reference "Sudarshi Premawardhana et al., 2020" has been added.

My final thoughts are it would be nice to discover if the S100A1 binding site is as extensive/competes with the MARP1 site which extends to the I81 domain. Although perhaps this is beyond the scope of the manuscript.

We agree that this is interesting and will be explored in future work.

Reviewer #2 (Comments to the Authors):

1. The author's discussion analyzes the implications of S100A1 binding (Line 422+), where cooperative or potentially competitive binding are discussed. However, function is not discussed. Do the conformational changes potentially relate to changes in muscle passive tension or other titin-related physiologic functions?

Titin plays a crucial role in maintaining muscle elasticity and passive tension. Its structural changes can affect the stiffness of muscle fibers. If S100A1 binds to titin, in regions such as PEVK known for modulating passive tension, it could potentially alter titin's elasticity. The same effect may occur if S100A1 binds to a region like UN2A.

2. *Binding Lifetime/Kinetics: The authors suggest that the weak binding may be helpful in a physiological context, and potentially differential by tissues. Since a major difference in muscle activation is the twitch activation of cardiac cells versus the twitch and possible tetanic activation of the skeletal muscles, would the binding lifetime be a significant physiologic factor?*

The binding lifetime would play a similar physiological role in cardiac muscle and skeletal muscle. In cardiac muscle, a shorter binding lifetime may facilitate quick response and release, ensuring efficient and timely contractions that are necessary for cardiac function when it undergoes twitch activation. In skeletal muscle, a more adaptable binding profile can support both rapid responses and sustained contractions, since it can undergo both twitch and tetanic activation.

3. *Methods- Plasmids:*

3a. *The plasmid sequences should be included in a supplement or reference for reproducibility.*

We've added the names of the host plasmids to the supplementary page.

3b. *It is unclear from the description of the UN2A whether flanking Ig domains may be included.*

We've added language to clarify that there are not any Ig domains included in our expression constructs.

4. *Results (Lines 220-222): The results describe a sequence analysis of the UN2A region and binding homology. This result may need to be described in the methods and the homology shown for rigor and reproducibility purposes. A reference to a previously published comparison between S100A1 and the UN2A region would be sufficient.*

We have included the consensus sequence of the S100A1 binding sequence into the manuscript as well as an alignment between the UN2A sequence and the binding sequence for the RyR receptor to help clarify the analysis we conducted to try to identify the binding sequence.

Reviewer #3

There is quite a bit of discussion on sequence analysis for S100A1 binding in the result section. I would like to see this analysis that compares the RYR1/2 S100A1 binding sequence to titin's N2A and PEVK sequence. Perhaps comparing among species can yield some further insights. Or if interesting, comparisons of CaM or Casq binding-sites.

We have a consensus sequence for S100A1 and did an alignment with the RyR sequence that binds to S100A1 and UN2A and yielded a 33.3% similarity in sequences. It has been added to the supplementary page.

I miss a potential role for S100A1 interacting with N2A. As interaction with fActin is frequently discussed, I'd like to see fActin co-sedimentation assays with UN2A and S100A1 at physiological pH and pH 6. Ideally I'd also like to see these experiments with mono- and di-calcium bound S100A1.

These are all interesting points that we are exploring but they are beyond the scope of this current study.

The UN2A is also known to be phosphorylated by PKA (Lanzicher et al 2020), does this phospho-site affect S100A1 binding? Your FRET assay seems ideal for probing the role.

This is an interesting point and we intend to do these studies but they are beyond the scope of this current work.

Do larger N2A constructs also interact the same with S100A1? UN2A-I81 or I80-UN2A might have more structural stability and bind S100A1 under more physiological conditions.

We are planning to perform these tests in future studies but have not currently developed expression clones for these constructs yet.

Your discussion would greatly benefit from moving away from physiological conditions for the S100A1-N2A interaction. The relatively extreme conditions needed for S100A1 to bind to N2A make this mechanism unlikely under normal/healthy conditions. Focusing on extreme exercise and muscle damage makes more sense in the context of your data.

We appreciate this suggestion but would argue that this binding interaction might occur under normal conditions, just at a low level. As the muscle moves to more extreme conditions, this binding will become stronger. We have tried to adjust the manuscript to make this more clear.

In addition, how would the broader S100 family of proteins contribute and are they likely to bind to N2A?

We have not tested other members of the S100 family yet but we intend to explore this in the future. We initiated these studies since S100A1 is predominantly found in muscle and is linked to other muscle proteins, which is why we started there. There is not a lot of sequence similarity between other members of the family, which is why we have not tested other family members to date.

Line 44; I suggest removing the approximate size at 4.0 mDa. Titin can range anywhere from 0.6-4.0 mDa depending on splicing.

We have removed this statement.

Line 46; Titin is an important contributor of passive tension to muscle. Describing this as passive force enhancement is confusing and I advise rewriting this sentence.

Thank you for this suggestion and we have revised this statement to make this clearer.

Line 65-66, there is a new structure for N2A released recently by Stehle et al

We have added the Stehle reference to the manuscript.

Line 68-69; MARP is a family of 3 proteins, see Miller et al 2003.

We have revised this line in the manuscript.

Line 75-76; N2A interacting with Ca²⁺ has been proposed by several groups. Interaction with fActin-N2A has been published by several groups, with the role of Ca²⁺ being ambiguous, some showing enhancement others showing no effect.

We acknowledge that the role of Ca²⁺ in potential binding of N2A to actin is still unclear. However, our published data suggests a role for calcium in this process and we are merely stating in this sentence that this was part of why we were exploring the potential for an interaction with S100A1 and N2A.

Line 121; the UN2A construct requires more description it is unclear if this is just the coiled-coil or the whole unique sequence.

We have included the amino acid range used for the UN2A expression clone, the sequence of the clone and have altered the description of the clone to make it clear that this included the whole sequence between I80 and I81.

Line 137; What was the source for the S100A1 plasmid, and which sequence was used.

We have included information about the S100A1 expression plasmid to the methods section.

Although I appreciate single graph pages for reviewing the manuscript, I do think the graphs can be condensed and combined into fewer figures to aid the narrative.

We have revised our figures to condense them into multi-panel figures where possible.

Line 333; same comment as in the introduction, several groups have looked at this interaction between N2A and fActin. MARP1 also seems to modify N2A-fActin binding in a calcium based mechanism (Zhou et al 2021 and Van Der Pijl et al 2021). Considering the MARP proteins possess Casq-binding sites, have you considered interaction of the MARPs and S100A1?

We have added additional citations to the manuscript regarding the MARP interactions. The potential of MARP and S100A1 interactions is intriguing and something that we plan to interrogate in the future but have tested this to date.

Line 336; titin kinase is not dependent on CaM, as shown by Lange 2005 Nature. The predicted CaM site in the kinase domain currently has no known relevance and even the kinase itself has disputed activity.

Thank you for this clarification and we have revised this in the manuscript.

There are several papers on the role of Ca²⁺ interacting with the PEVK segment of titin that should be cited and discussed here.

We have added references for calcium interacting with the PEVK as requested.

Line 353/354; Please provide citations for the physiological pH and pCa in striated muscle.

We have provided citations for physiological pH and pCa in striated muscle.

Line 374; What did your SEC data show as far as mono-, di- or oligomeric binding of S100A1 to UN2A? Did calcium concentration change the conformation of UN2A by itself (neutral pH)?

We have added SEC data for the UN2A by itself, but we do not see any changes to the UN2A at high calcium concentrations.

Line 406; Extended muscle activation only minimally increases the pool of intracellular calcium and tetanic contraction rarely happens in skeletal muscle.

We apologize for any confusion, but our proposed model is not based on tetanic contractions but rather a response based on a repeated activation of the muscle and have revised the manuscript to reflect this.

Line 424-426; This segment is poorly written and misdescribes the functions of the N2A-binding proteins: MARPs are not known as transcription factors. MARPs are adapter/chaperone proteins that can interact with transcription factors. Smyd2 is a methyltransferase that can methylate HSP90ab, a PTM which makes Hsp90 bind to N2A. I suggest looking up several recent reviews on N2A and interacting proteins.

We have revised this section to include Voelkel et al (2013) and Donlin et al (2012) to make this point clearer.

August 6, 2024

Dr. Matthew J Gage
University of Massachusetts Lowell
One University Avenue
Lowell, MA 01854

Re: 202313472R1

Dear Dr. Gage,

Thank you for submitting your manuscript, entitled "S100A1 Binds to Titin's N2A Insertion Sequence in a pH Dependent Manner" to JGP. Your manuscript has now been seen by 3 reviewers, whose comments are appended below. You will see that the reviewers are very interested in your work but also raised several concerns that should be addressed prior to further consideration of the manuscript at JGP. In particular, confirming with an alternative method N2A-S100A1 binding, or at a minimum showing the composition of the proteins contained in the chromatograph peaks of the S100 N2A mixture, in the presence of Ca²⁺ (Fig 2A).

We would be pleased to receive a suitably revised manuscript that addresses these concerns, which will be re-reviewed, most likely by some or all of the original referees. In addition, please do not hesitate to contact me (via the editorial office) if you feel that a discussion of the reviewers' and editors' comments would be helpful.

Please submit your revised manuscript via the link below along with a point-by-point letter that details your responses to the editors' and reviewers' comments, as well as a copy of the text with alterations highlighted (boldfaced or underlined). If the article is eventually accepted, it would include a 'revised date' as well as submitted and accepted dates. If we do not receive the revised manuscript within one year, we will regard the article as having been withdrawn. We would be willing to receive a revision of the manuscript at a later time, but the manuscript will then be treated as a new submission, with a new manuscript number.

Please pay particular attention to recent changes to our instructions to authors in the following sections: Data presentation, Blinding and randomization and Statistical analysis, under Materials and Methods, as shown here: <https://rupress.org/jgp/pages/submission-guidelines#prepare>. Re-review will be contingent on inclusion of the required information (including for data added during revision) and demonstration of the experimental reproducibility of the results. Also, to improve the reproducibility of published content, we have partnered with SciScore. Authors are prompted in eJP to copy and paste the Materials and Methods section of their manuscript for a SciScore assessment when submitting their revised manuscript. Authors are encouraged (not required) to further revise their Materials and Methods if the SciScore is below 4. More information can be found here: <https://rupress.org/jgp/pages/submission-guidelines#sciscore>

Please note, JGP now requires authors to submit Source Data used to generate figures containing gels and Western blots with all revised manuscripts (when applicable). This Source Data consists of fully uncropped and unprocessed images for each gel/blot displayed in the main and supplemental figures. If your paper includes cropped gel and/or blot images, please be sure to provide one Source Data file for each figure that contains gels and/or blots along with your revised manuscript files. File names for Source Data figures should be alphanumeric without any spaces or special characters (i.e., SourceDataF#, where F# refers to the associated main figure number or SourceDataFS# for those associated with Supplementary figures). The lanes of the gels/blots should be labeled as they are in the associated figure, the place where cropping was applied should be marked (with a box), and molecular weight/size standards should be labeled wherever possible.

Source Data files will be made available to reviewers during evaluation of revised manuscripts and, if your paper is eventually published in JGP, the files will be directly linked to specific figures in the published article.

Source Data Figures should be provided as individual PDF files (one file per figure). Authors should endeavor to retain a minimum resolution of 300 dpi or pixels per inch. Please review our instructions for export from Photoshop, Illustrator, and PowerPoint here: <https://rupress.org/jgp/pages/submission-guidelines#revised>

When revising your manuscript, please be sure it is a double-spaced MS Word file and that it includes editable tables, if appropriate.

Please submit your revised manuscript via this link:
Link Not Available

Thank you for the opportunity to consider your manuscript.

Sincerely,

Henk L. Granzier, Ph.D.
On behalf of Journal of General Physiology

Journal of General Physiology's mission is to publish mechanistic and quantitative molecular and cellular physiology of the highest quality; to provide a best-in-class author experience; and to nurture future generations of independent researchers.

Reviewer #1 (Comments to the Authors):

The GF is now more convincing, the lack of SDS PAGE to accompany it is, however, disappointing and means the experiment is not as robust as it could be - I am personally not 100% convinced about the binding. If this experiment cannot be completed (perhaps with a collaborator with the necessary equipment?) I would still prefer to see a second method such as a pull-down - as this experiment is not technically difficult to achieve and can be done even in a resource-poor lab.

The alignment should be in the main figures not just in the supplementary. I cannot say it is very convincing to me, but if included in the main manuscript others can judge quickly for themselves.

I am satisfied with the protein's purity for the described experiments and the SPR non-specific binding control. This part of the manuscript is to a much higher standard now.

Minor points:

MARP1 was shown to facilitate N2A binding to F-actin, causing an increase in passive force and constraining overstretching of sarcomeres. (Van Der Pijl et al., 2021)

Missing reference Zhou et al 2021

Fig2 from the figure legend what the inset graph is showing is unclear, is this different calcium condition?

Fig4 label for clarity please add what uM refers to on the figure

Fig3C label for clarity please add what x axis conc refers to

There are still no details of the plasmids used (although the reply says that this detail was added?) which doesn't allow for a full assessment of the purification procedure or what cloning artifacts could be left on the proteins (which is important to assess the binding experiments) tags present or cleavage sites. I'm assuming due to the IEX method there were no affinity tags but this needs to be detailed to be sure. This goes for both proteins.

Reviewer #2 (Comments to the Authors):

The authors of Manuscript # 202313472R1, "S100A1 Binds to Titin's N2A Insertion Sequence in a pH Dependent Manner", have revised the manuscript and addressed this reviewer's original concerns.

This reviewer does suggest adding a panel to Figure 1 showing either a new panel with the estimated conformation of UN2A and S100A1 after binding or an arrow in Figure 1B indicating the shift that would occur (increasing distance) that would relate to the data reported in the manuscript.

Reviewer #3 (Comments to the Authors):

I once again congratulate the authors on their S100A1-N2A manuscript. This work provides new insights into calcium regulation of titin's N2A element, which is an important regulator in striated muscle. The revised manuscript by Apel et al has been improved and after reviewing the manuscript, I have listed several suggestions below that would benefit the manuscript.

Minor comments:

The method section has a great deal of repetition in how the proteins were expressed a more summarized version with detail for individual proteins will greatly reduce the repetitive text.

Sub-figures are not always cited in the result section, example fig. 6b

There is some progress in condensing figures, but there is still room for improvement. Example fig 1-3 could easily be combined as they all support the same concept of UN2A-S100A1 interaction. It would greatly improve the readability of the manuscript if the figures highlight the key findings; interaction, Ca-sensitivity, pH etc.

Major comments:

As reviewer 1 mentioned, the retention time difference between the bound and unbound states of S100A1 to UN2A is not very convincing. A secondary verification using a native gel or co-precipitation would strengthen the proposed interaction. Immobilizing the FRET construct on a His-trap and subsequently flowing S100A1 through +/- Ca²⁺ can address this. That will also circumvent the size issue of the constructs and provide insight into S100A1 binding as a homodimer or just a single molecule.

I like the proposed mechanism for inhibiting fActin binding, but why not show it. A simple fActin Co-sedimentation assay +/- Ca²⁺ and +/- S100A1 is an easy way to demonstrate this hypothesis.

The N2A conformational change in the FRET assay is rather interesting. Does a pCa 6-7 show an intermediary shift? If possible, looking at the conformational change with time resolved FRET and a stopped flow system to alter Ca²⁺ concentrations could be very interesting.

We appreciate the additional careful and critical feedback on our revised manuscript. A point by point description of how we addressed the reviewer comments is listed below.

Reviewer #1:

The GF is now more convincing, the lack of SDS PAGE to accompany it is, however, disappointing and means the experiment is not as robust as it could be - I am personally not 100% convinced about the binding. If this experiment cannot be completed (perhaps with a collaborator with the necessary equipment?) I would still prefer to see a second method such as a pull-down - as this experiment is not technically difficult to achieve and can be done even in a resource-poor lab.

Thank you for this suggestion and we have included a Co-IP experiment to further support our HPLC results. This shows S100A1 capture of UN2A, which we believe that, combined with the observed altered migration of the complex samples in the presence but not absence of Ca^{2+} , the SPR and FRET data makes a compelling case that the S100A1 binds to this region of titin. We acknowledge that the shifts in the HPLC are not large and the amount of UN2A captured in the Co-IP is lower than the S100A1. However, as our SPR data demonstrates, this interaction has a μM binding affinity, so large shifts and equimolar binding would not be expected.

The alignment should be in the main figures not just in the supplementary. I cannot say it is very convincing to me, but if included in the main manuscript others can judge quickly for themselves.

We appreciate your suggestion but respectfully disagree that the alignment should be included in the main manuscript. This alignment was added in response to a reviewer comment from our initial submission and is intended only to highlight that there is low to no homology between the known S100A1 binding sequence in the RyR receptor and the sequence of the UN2A region rather than to suggest that the two binding sites are similar. Given that other reviewers have asked us to consolidate figures and that this is a minor point, we have tried to highlight that there is no similarity in the text and believe that this should remain in the supplemental information.

Minor points:

MARP1 was shown to facilitate N2A binding to F-actin, causing an increase in passive force and constraining overstretching of sarcomeres. (Van Der Pijl et al., 2021)

Missing reference Zhou et al 2021

We have included the reference, Zhou et al 2021.

Fig2 from the figure legend what the inset graph is showing is unclear, is this different calcium condition?

We have removed the inset graph from figure 2.

Fig4 label for clarity please add what uM refers to on the figure

We have updated Figure 4 to clarify that ' μM ' refers to the concentration of S100A1.

There are still no details of the plasmids used (although the reply says that this detail was added?) which doesn't allow for a full assessment of the purification procedure or what cloning artifacts could be left on the proteins (which is important to assess the binding experiments) tags present or cleavage sites. I'm assuming due to the IEX method there were no affinity tags but this needs to be detailed to be sure. This goes for both proteins.

The details of the plasmids used, including any tags and cleavage sites, are provided in the supplementary data.

Reviewer #2 (Comments to the Authors):

This reviewer does suggest adding a panel to Figure 1 showing either a new panel with the estimated conformation of UN2A and S100A1 after binding or an arrow in Figure 1B indicating the shift that would occur (increasing distance) that would relate to the data reported in the manuscript.

Thank you for your suggestion about revising Figure 1 to indicate the changes in UN2A that might occur with S100A1 binding. However, we currently do not know the specific S100A1 binding site, making any modeling of the potential conformational change speculative at best. While the FRET data indicates a change in the conformation, we don't know if this change involves a reorganization of the tri-helix bundle, reorganization of the disordered regions or some combination of both. We feel that providing any sort of suggestion of the conformational shifts in UN2A from binding would suggest a higher level of confidence that would be justified. However, we are working to try to understand this shift and hope to be able to report that in the future.

Reviewer #3 (Comments to the Authors):

The method section has a great deal of repetition in how the proteins were expressed a more summarized version with detail for individual proteins will greatly reduce the repetitive text.

Each protein in our study has a unique expression and purification profile. These differences are critical for the reproducibility of our work. Therefore, we have provided detailed methods for each protein to ensure that future researchers can accurately replicate our experiments.

Sub-figures are not always cited in the result section, example fig. 6b

Thank you for your observation. We have reviewed the manuscript and ensured that all sub-figures, including Figure 6B, are properly cited in the Results section.

There is some progress in condensing figures, but there is still room for improvement. Example fig 1-3 could easily be combined as they all support the same concept of UN2A-S100A1 interaction. It would greatly improve the readability of the manuscript if the figures highlight the key findings; interaction, Ca-sensitivity, pH etc.

We appreciate the suggestion about how to condense figures and have combined a couple additional figures. However, we feel that the current layout of the figures does highlight the key findings and fits the way that the data is presented. Further consolidation of figures would require a major revision of the text or figure panels being significantly separated from the text where the results are described, which we feel would decrease the readability of the manuscript.

Major comments:

As reviewer 1 mentioned, the retention time difference between the bound and unbound states of S100A1 to UN2A is not very convincing. A secondary verification using a native gel or co-precipitation would strengthen the proposed interaction. Immobilizing the FRET construct on a His-trap and subsequently flowing S100A1 through +/- Ca²⁺ can address this. That will also circumvent the size issue of the constructs and provide insight into S100A1 binding as a homodimer or just a single molecule.

Thank you for the suggestion. We agree that additional verification of the interaction would be beneficial. We have conducted a co-immunoprecipitation experiment, which we have now included in the manuscript as well.

I like the proposed mechanism for inhibiting fActin binding, but why not show it. A simple fActin Co-sedimentation assay +/- Ca²⁺ and +/- S100A1 is an easy way to demonstrate this hypothesis.

Thank you for the suggestion regarding the f-actin co-sedimentation assay. We agree that this experiment could provide valuable insights into the proposed mechanism for inhibiting f-actin binding. However, due to the difficulties of the expression and solubility of the complete N2A region, which is necessary to test this model, we are unable to perform this assay at this time. We are working to address this expression issue so we can further test this model.

The N2A conformational change in the FRET assay is rather interesting. Does a pCa 6-7 show an intermediary shift? If possible, looking at the conformational change with time resolved FRET and a stopped flow system to alter Ca²⁺ concentrations could be very interesting.

Thank you for your interest in the N2A conformational change observed in the FRET assay. We have not performed time-resolved FRET or stopped-flow experiments to investigate the conformational change with varying calcium concentrations yet but we appreciate your suggestion and are considering these experiments as we further dissect the nature of this interaction.

November 6, 2024

Dr. Matthew J Gage
University of Massachusetts Lowell
One University Avenue
Lowell, MA 01854

Re: 202313472R2

Dear Matt,

Thank you for submitting your manuscript, entitled "S100A1 Binds to Titin's N2A Insertion Sequence in a pH Dependent Manner" to JGP. Your manuscript has now been seen by 2 reviewers, whose comments are appended below. You will see that the reviewers are enthusiastic about the study and its potential impact and raised only a minor concern that should nevertheless be addressed before further consideration of the manuscript at JGP. In particular, the ratio of S100a-N2A binding in the Co-IP remains to be determined.

We hope that you will be able to submit a revised manuscript that addresses this point, which we believe will pose no problems and which may or may not be re-reviewed. In addition, please do not hesitate to contact me (via the editorial office) if you feel that a discussion of the reviewers' and editors' comments would be helpful.

Please submit your revised manuscript via the link below, along with a point-by-point letter that details your response to the reviewers' and editors' comments, as well as a copy of the text with alterations highlighted (boldfaced or underlined). If the article is eventually accepted, it would include a 'revised date' as well as submitted and accepted dates. If we do not receive the revised manuscript within one year, we will regard the article as having been withdrawn. We would be willing to receive a revision of the manuscript at a later time, but the manuscript will then be treated as a new submission, with a new manuscript number.

Please pay particular attention to recent changes to our instructions to authors in the following sections: Data presentation, Blinding and randomization and Statistical analysis, under Materials and Methods, as shown here: <https://rupress.org/jgp/pages/submission-guidelines#prepare>. Re-review will be contingent on inclusion of the required information (including for data added during revision) and demonstration of the experimental reproducibility of the results. Also, To improve the reproducibility of published content, we have partnered with SciScore. Authors are prompted in eJP to copy and paste the Materials and Methods section of their manuscript for a SciScore assessment when submitting their revised manuscript. Authors are encouraged (not required) to further revise their Materials and Methods if the SciScore is below 4. More information can be found here: <https://rupress.org/jgp/pages/submission-guidelines#sciscore>.

Please note, JGP now requires authors to submit Source Data used to generate figures containing gels and Western blots with all revised manuscripts (when applicable). This Source Data consists of fully uncropped and unprocessed images for each gel/blot displayed in the main and supplemental figures. If your paper includes cropped gel and/or blot images, please be sure to provide one Source Data file for each figure that contains gels and/or blots along with your revised manuscript files. File names for Source Data figures should be alphanumeric without any spaces or special characters (i.e., SourceDataF#, where F# refers to the associated main figure number or SourceDataFS# for those associated with Supplementary figures). The lanes of the gels/blots should be labeled as they are in the associated figure, the place where cropping was applied should be marked (with a box), and molecular weight/size standards should be labeled wherever possible.

Source Data files will be made available to reviewers during evaluation of revised manuscripts and, if your paper is eventually published in JGP, the files will be directly linked to specific figures in the published article.

Source Data Figures should be provided as individual PDF files (one file per figure). Authors should endeavor to retain a minimum resolution of 300 dpi or pixels per inch. Please review our instructions for export from Photoshop, Illustrator, and PowerPoint here: <https://rupress.org/jgp/pages/submission-guidelines#revised>

Whilst you are revising your manuscript, we ask that you consider whether you have any artwork that might be suitable for the cover of JGP. Microscopy images are particularly good for cover artwork, but other types of image can be very effective, so we encourage you to be creative. Please don't restrict yourself to images from the paper; an image that is relevant to the work described would be just as suitable. Images should be a minimum resolution of 300 dpi. To see recent examples, visit the following page and click on 'Show covers? Yes': <https://jgp.rupress.org/content/by/year>

Thank you for submitting your interesting research to JGP.

Please submit your revised manuscript, and any associated files, via this link:

Link Not Available

Sincerely,

Henk L. Granzier, Ph.D.

On behalf of Journal of General Physiology

Journal of General Physiology's mission is to publish mechanistic and quantitative molecular and cellular physiology of the highest quality; to provide a best-in-class author experience; and to nurture future generations of independent researchers.

Reviewer #2 (Comments to the Authors):

(No Additional Comments)

Reviewer #3 (Comments to the Authors):

I once again congratulate the authors on their interesting work on S100a-N2A interaction. I only have a minor comment: What is the ratio of S100a-N2A binding in the Co-IP? More S100a binding to N2A could indicate multiple binding sites or multiple S100a molecules binding to a single binding site can alter strength of interaction with fActin. A brief paragraph in the discussion section would provide some perspective.

We appreciate the additional careful and critical feedback on our revised manuscript. A point-by-point description of how we addressed the reviewer comments is listed below.

Reviewer #3:

I once again congratulate the authors on their interesting work on S100a-N2A interaction. I only have a minor comment: What is the ratio of S100a-N2A binding in the Co-IP? More S100a binding to N2A could indicate multiple binding sites or multiple S100a molecules binding to a single binding site can alter strength of interaction with fActin. A brief paragraph in the discussion section would provide some perspective.

Thank you for the feedback. In response, we have conducted an ImageJ analysis to quantify the binding ratio and included this information in the manuscript, along with the molar ratio used in the Co-IP assay.

While our model proposes that S100A1 binding to N2A could potentially interfere with N2A's interaction with f-Actin, this is currently speculative and there is not experimental data to support this model. We have sections discussing the significance of the weak binding interactions and feel that adding more discussion about a theoretical model could suggest more confidence in this model than is currently warranted.

November 18, 2024

Dr. Matthew J Gage
University of Massachusetts Lowell
One University Avenue
Lowell, MA 01854

Re: 202313472R3

Dear Dr. Gage,

I am pleased to let you know that your manuscript, entitled "S100A1 Binds to Titin's N2A Insertion Sequence in a pH Dependent Manner" is scientifically acceptable for publication in Journal of General Physiology. Formal acceptance will follow when it is modified in accordance with our editorial policies.

Please note items that need attention are listed at the bottom of this email (under 'manuscript formatting checklist') and on the attached marked-up pdf file. Please also be sure to include a letter addressing the reviewers' comments point-by-point (if applicable) and a copy of the text with alterations highlighted (boldfaced or underlined). Your manuscript should be a double-spaced MS Word file and include editable tables, if appropriate.

JGP requires a data availability statement for all research article submissions. These statements will be published in the article directly above the Acknowledgments. The statement should address all data underlying the research presented in the manuscript. Please visit the JGP instructions for authors for guidelines and examples of statements at <https://rupress.org/jgp/pages/editorial-policies#data-availability-statement>.

Lastly, JGP adds short captions to articles listed on our weekly newest article emails. If you haven't, please provide a short, ~40-word summary statement for the online JGP table of contents and alerts. This summary should describe the context and significance of the findings for a general readership and be placed on/near the title page.

Please submit your final files via this link:
Link Not Available

Thank you for choosing to publish your research in JGP and please feel free to contact me with any questions.

Sincerely,

Henk L. Granzier, Ph.D.
On behalf of Journal of General Physiology

Journal of General Physiology's mission is to publish mechanistic and quantitative molecular and cellular physiology of the highest quality; to provide a best in class author experience; and to nurture future generations of independent researchers.

Manuscript formatting checklist:

- MS Word document of text needed (including editable tables)
- MS Word document of supplemental text needed, if applicable (including figure legends and editable tables)
- Brief Statement describing supplementary information needed, if applicable (in subsection at end of Materials & Methods)
- Please include a data availability statement preceding the Acknowledgments section. Please see <https://rupress.org/jgp/pages/editorial-policies#data-availability-statement>
- Figures created at sufficient resolution and in acceptable format (including supplemental if applicable). If working in Illustrator, we prefer .ai or .eps file format. If working in Photoshop please use 600dpi/1000dpi .tiff or .psd file format. Minimum resolution at estimated print size: Minimum resolution for all figures is 600 dpi. For figures that contain both photographs and line art or text, 600 dpi is highly recommended. Figures containing only black and white elements (line art, no color, and no gray) should be 1,000 dpi. Maximum figure size is 7 in wide x 9 in high (17.5 x 22.8 cm) at the correct resolution. <https://jgp.rupress.org/fig-vid-guidelines>

- Supplemental figures, if any, conforming to same guidelines as manuscript figures (noted above)
 - If images resemble one from a prior publications, the author must seek permissions (to reproduce or adapt) from the original publisher. [You can resubmit your paper while waiting to hear back from the original publisher but please keep us updated]
 - All authors must complete a disclosure form prior to acceptance. A link to complete the form has been sent to all coauthors. Please provide the editorial office with updated email addresses if necessary
-

Reviewer #2 (Comments to the Authors):

(No additional Comments)

Reviewer #3 (Comments to the Authors):

No further comments.